# Oceanographic regional climate projections for the Baltic Sea until 2100

H. E. Markus Meier[1,2], Christian Dieterich[2,+], Matthias Gröger[1], Cyril Dutheil[1], Florian Börgel[1], Kseniia Safonova[1], Ole B. Christensen[3] and Erik Kjellström[2]

[1]Department of Physical Oceanography and Instrumentation, Leibniz Institute for Baltic Sea Research Warnemünde, Rostock, 18119, Germany
[2]Research and Development Department, Swedish Meteorological and Hydrological Institute, Norrköping, 601 76, Sweden
[3]Danish Climate Center, Danish Meteorological Institute, Copenhagen, Denmark
[+]Deceased

*Correspondence to*: H.E. Markus Meier (markus.meier@io-warnemuende.de)

**Abstract.** The Baltic Sea, located in northern Europe, is a semi-enclosed, shallow and tide-less sea with seasonal sea-ice cover in its northern sub-basins. Its long water residence time contributes to oxygen depletion in the bottom water of its southern sub-basins. In this study, recently performed scenario simulations for the Baltic Sea including marine biogeochemistry were analysed and compared with earlier published projections. Specifically, dynamical downscaling using a regionally coupled atmosphere-ocean climate model was used to regionalise four global Earth System Models. However, as the regional climate model does not include components representing terrestrial and marine biogeochemistry, an additional catchment and a coupled physical-biogeochemical model for the Baltic Sea were included. Previous scenario simulations and scenarios taking into account the impact of various water levels were examined. According to the projections, compared to the present climate, higher water temperatures, a shallower mixed layer with a sharper thermocline during summer, less sea-ice cover and greater mixing in the northern Baltic Sea during winter can be expected. Both the frequency and the duration of marine heat waves will increase significantly, in particular in the coastal zone of the southern Baltic Sea (except in regions with frequent upwellings). Nonetheless, due to the uncertainties in the projections regarding regional winds, the water cycle and the global sea level rise, robust and statistically significant salinity changes could not be identified. The impact of a changing climate on biogeochemical cycling is predicted to be considerable but still smaller than that of plausible nutrient input changes. Implementing the proposed Baltic Sea Action Plan, a nutrient input abatement plan for the entire catchment area, would result in a significantly improved ecological status of the Baltic Sea, including reductions in the size of the hypoxic area also in a future climate, which in turn would increase the resilience of the Baltic Sea against anticipated climate change. While our findings regarding changes in heat-cycle variables mainly confirm earlier scenario simulations, they differ substantially from earlier projections of salinity and biogeochemical cycles, due to differences in experimental setups and in input scenarios for bioavailable nutrients.

## 1 Introduction

The Baltic Sea is a shallow, semi-enclosed sea located in northern Europe (Fig. 1). It has a mean depth of 54 m but due to its strongly varying bottom topography it can be divided into several sub-basins, with limited transport between them (Sjöberg, 1992). In particular, water exchange between the Baltic Sea and the North Sea is hampered

because of two shallow sills located in narrow channels connecting these two water bodies. Thus, large saltwater inflows occur only sporadically, on average once per year, mainly during the winter season but never during summer (Mohrholz, 2018). Furthermore, because the Baltic Sea is embedded within a catchment area that is about four times larger than the Baltic Sea surface, annual freshwater inputs are large relative to the volume of the Baltic Sea (Bergström and Carlsson, 1994). The volume of the Baltic Sea is ~21,700 km$^3$ (Sjöberg, 1992) and the turnover time of the total freshwater supply (~16,000 m$^3$ s$^{-1}$) is 35 years (Meier and Kauker, 2003). These features contribute to strong horizontal and vertical salinity gradients in the Balti Sea (Fonselius and Valderrama, 2003). Moreover, due to its location and physical characteristics, especially the long water-residence time, the Baltic Sea is vulnerable to external pressures, including eutrophication, pollution and global warming (e.g., Jutterström et al., 2014). Ocean circulation modelling has shown that the time scale of the salinity response to changes in atmospheric and hydrological forcing is 20 years (Meier, 2006).

Some 85 million people, in 14 countries, currently live in the catchment area of the Baltic Sea, and anthropogenic pressure on the marine ecosystem is accordingly high (HELCOM, 2018). Insufficiently treated wastewater, pollutant emissions, overfishing, habitat degradation and intensive marine traffic, including oil transport, place a heavy burden on the Baltic Sea ecosystem (Reckermann et al., 2021). One consequence is oxygen depletion of the Baltic Sea's deep waters, such that bottom areas lack higher life forms (e.g., Carstensen et al., 2014; Meier et al., 2018b). In 2018, the area of dead bottom was equal to that of the Republic of Ireland, ~73,000 km$^2$, which is about one sixth of the sea surface area of the Baltic Sea. Oxygen depletion in the deeper parts of the Baltic Sea arise from the limited ventilation of those waters and the accelerated oxygen consumption that accompanies the remineralisation of organic matter (Meier et al., 2018b). Hence, nutrient input abatement strategies, such as the Baltic Sea Action Plan (BSAP), have been proposed (HELCOM, 2007), with projections of their impact requested by stakeholders such as the Helsinki Commission (HELCOM) or national environmental protection agencies[1].

Projections of the Baltic Sea's climate at the end of the 21[st] century were among the first to be made for coastal seas worldwide (Meier and Saraiva, 2020). Already at the beginning of the 2000s, the first scenario simulations, were carried out for selected time slices in present and future climates (e.g. Haapala et al., 2001; Meier, 2002a, b; Omstedt et al., 2000). In the dynamical downscaling approach used for those simulations, regional climate models (RCMs) were employed to refine predictions of global climate change to regional and local scales, in this case for the Baltic Sea (e.g. Rummukainen et al., 2004; Döscher et al., 2002). However, those first projections were based on scenarios consisting of a single global climate model (GCM) and a single greenhouse gas (GHG) concentration (150% increase in equivalent $CO_2$ concentration in the atmosphere in the future vs. the historical climate) and only covered 10-year time slices. These initial attempts were therefore followed by more advanced scenario simulations using mini-ensembles (e.g. Döscher and Meier, 2004; Meier et al., 2004b; Meier et al., 2004a; Räisänen et al., 2004) and centennial-long simulations (e.g. Meier, 2006; Meier et al., 2006; Meier et al., 2011c; Table 1). However, the latter studies considered only monthly mean changes in the future vs. the present climate, applying a so-called delta approach, while neglecting possible changes in inter-annual variability. From these oceanographic studies it was concluded that "mean annual sea surface temperatures (SSTs) could increase by some 2 to 4˚C by the end of the 21st century. Ice extent in the sea would then decrease by some 50 to 80%. The average salinity of

[1]https://helcom.fi/helcom-at-work/events/events-2021/ccfs-launch/

the Baltic Sea could range between present day values and decreases of as much as 45%. However, it should be
noted that these oceanographic findings, with the exception of salinity, are based upon only four regional scenario
simulations using two emissions scenarios and two global models" (BACC Author Team, 2008).
For the second assessment of climate change in the Baltic Sea region (BACC II Author Team, 2015), continuously
integrated transient simulations from present to future climates became available and even included marine
biogeochemical modules (e.g. Eilola et al., 2013; Friedland et al., 2012; Gräwe and Burchard, 2012; Gräwe et al.,
2013; Gröger et al., 2019; Gröger et al., 2021b; Holt et al., 2016; Kuznetsov and Neumann, 2013; Meier et al.,
2011b; Meier et al., 2011c; Meier et al., 2012a; Meier et al., 2012c; Meier et al., 2012d; Neumann, 2010; Neumann
et al., 2012; Omstedt et al., 2012; Pushpadas et al., 2015; Ryabchenko et al., 2016; Skogen et al., 2014) and higher
trophic levels (e.g. Bauer et al., 2019; Ehrnsten et al., 2020; Gogina et al., 2020; Holopainen et al., 2016;
MacKenzie et al., 2012; Niiranen et al., 2013; Vuorinen et al., 2015; Weigel et al., 2015). The BACC II Author
Team (2015) concluded that "recent studies confirm the findings of the first assessment of climate change in the
Baltic Sea basin". A key finding of their report was that "No clear tendencies in saltwater transport were found.
However, the uncertainty in salinity projections is likely to be large due to biases in atmospheric and hydrological
models. Although wind speed is projected to increase over sea, especially over areas with diminishing ice cover,
no significant trend was found in potential energy …" (a measure of energy to homogenize the water column). "In
accordance with earlier results, it was found that sea-level rise has greater potential to increase surge levels in the
Baltic Sea than does increased wind speed. In contrast to the first BACC assessment (BACC Author Team, 2008),
the findings reported in this chapter are based on multi-model ensemble scenario simulations using several GHG
emissions scenarios and Baltic Sea models. However, it is very likely that estimates of uncertainty caused by biases
in GCMs are still underestimated in most studies" (BACC II Author Team, 2015).
Since the early 21st century, transient simulations for the period 1960–2100 using regional ocean (Holt et al., 2016;
Pushpadas et al., 2015) and regionally coupled atmosphere-ocean models, so-called Regional Climate System
Models (RCSMs; Bülow et al., 2014; Dieterich et al., 2019; Gröger et al., 2019; Gröger et al., 2021b), have been
available for the entire combined Baltic Sea and North Sea system. An overview was given by Schrum et al. (2016)
as part of the North Sea Region Climate Change Assessment Report (NOSCCA, Quante and Colijn, 2016) and by
Gröger et al. (2021a) within the Baltic Earth Assessment Report (BEAR) project (this thematic issue).
There is a notable difference in the salinity projections between the first two assessments (BACC Author Team,
2008; BACC II Author Team, 2015) and recent scenario simulations (Meier et al., 2021). The first Baltic Sea
scenario simulations, driven by nine RCMs and five GCMs, showed a pronounced negative ensemble mean change
in salinity because two of the GCMs included a significant increase in the mean west wind component (Meier et
al., 2006). These pronounced changes in the large-scale atmospheric circulation were not a feature of later studies
(Saraiva et al., 2019a). However, as the natural variability was poorly sampled, this finding may be coincidental.
The large spread in river discharge did not decrease between the studies, ranging from −8% to +26% (Meier et al.,
2006; 2021). Since in more recent assessments the projected rates of global sea level rise (SLR) were revised
upwards (e.g. IPCC, 2019a; Bamber et al., 2019), recent scenario simulations for the Baltic Sea also considered a
rise in sea level (Meier et al., 2021). As a consequence of compensating effects of the competing drivers of salinity
changes, i.e. wind, freshwater input and sea level, future salinity changes were predicted to be small (Table 2).

In the following, we provide an overview of the projections performed since 2013, i.e. after the last assessment of
climate change for the Baltic Sea basin, and compare recent results with previous findings by the BACC II Author
Team (2015). We focus on projections for the marine environment, from both physical and biogeochemical
perspectives. Among the analysed variables are temperature, salinity, oxygen, phosphate, nitrate, phytoplankton,
primary production, nitrogen fixation, hypoxic area and Secchi depth (measuring water transparency). An
accompanying study by Christensen et al. (2021) investigated atmospheric projections in the Baltic Sea region.
For an overview of the development of RCSMs and their applications, the reader is referred to Gröger et al.
(2021a). In our comparisons of the various scenario simulations, we analyse only published data (Table 1), with a
focus on two recently generated sets of scenario simulations: BalticAPP and CLIMSEA (Table 1, see Saraiva et
al., 2019a, b; Meier et al., 2019a; 2021). These are compared with the previous ECOSUPPORT scenario
simulations (Meier et al., 2014) assessed by the BACC II Author Team (2015). Investigations of the impact of
climate change on primary production in the Baltic Sea that did not utilise a RCM (Holt et al., 2016; Pushpadas et
al., 2015) are not addressed herein, nor are nutrient input reduction scenarios under present climate, e.g. as
described by Friedland et al. (2021). To our knowledge, further coordinated experiments aimed at projections for
the coupled physical-biogeochemical system of the Baltic Sea after 2013 have not been published. Uncoordinated
scenario simulations performed prior to 2013 (including Ryabchenko et al., 2016) and their uncertainties were
previously discussed by Meier et al. (2018a; 2019b).

The paper is organised as follows. In Section 2, the dynamical downscaling method, the catchment and Baltic Sea
models, the experimental setup and the analytical strategy are introduced. In Section 3, the historical and future
climates results of the three scenario simulations, ECOSUPPORT, BalticAPP and CLIMSEA, are compared.
Tables 1, 3 and 4 provide an overview of these (Tables 3 and 4) and other (Table 1) scenario simulations from the
literature. A consideration of knowledge gaps and a summary of our findings conclude the study. Acronyms used
in this study are defined in Table 5.

**2 Methods**

**2.1 Regionalisation of a changing climate**

Dieterich et al. (2019) produced an ensemble of scenario simulations with a coupled RCSM, called RCA4-NEMO,
which was introduced by Wang et al. (2015). Gröger et al. (2019; 2021b) and Dieterich et al. (2019) validated and
analysed the different aspects of the RCA4-NEMO ensemble discussed herein. The atmospheric component,
RCA4 (Rossby Centre Atmosphere model Version 4), was run at a resolution of 0.22° and 40 vertical levels in the
EURO-CORDEX domain (Jacob et al., 2014), and the coupled North Sea-Baltic Sea model NEMO (Nucleus for
European Modelling of the Ocean) at a resolution of two nautical miles (3.7 km) and 56 levels. The two
components of the RCSM are coupled by sending sea surface data of sea level pressure, energy, mass and
momentum fluxes every 3 h from the atmosphere to the ocean model. Conversely, the atmosphere model receives
data of sea and ice surface temperatures and the sea-ice fraction and albedo at the same frequency.

This RCSM was applied to downscale eight different Earth System Models (ESMs), each one driven by three
Representative Concentration Pathways (RCPs). For the Baltic Sea projections, four ESMs (MPI-ESM-LR, EC-
Earth, IPSL-CM5A-MR, HadGEM2-ES; see Gröger et al. (2019) and references for the ESMs therein) and the
GHG concentration scenarios RCP4.5 and RCP8.5 were selected (Table 3). The four ESMs were part of the Fifth
Coupled Model Intercomparison Project (CMIP5; Taylor et al., 2012) and their results were assessed in the Fifth
IPCC Assessment Report (AR5; IPCC, 2013).

Surface variables of the atmospheric component were saved at hourly to 6-hourly frequencies to allow for an
analysis of means and extremes in present and future climates. As RCA4-NEMO does not contain model
components for terrestrial and marine biogeochemistry, two additional models forced with the atmospheric surface
fields of RCA4-NEMO, i.e. a catchment and a marine ecosystem model, were employed (Fig. 2).

For the ECOSUPPORT scenario simulations, dynamical downscaling was performed with the regional Rossby
Centre Atmosphere Ocean (RCAO) model (Döscher et al., 2002). RCAO consists of the atmospheric component
RCA3 (Samuelsson et al., 2011) and the oceanic component RCO (Meier et al., 2003; Meier, 2007), with horizontal
grid resolutions of 25 km and six nautical miles (11.1 km) respectively. In the vertical, the ocean model has 41
levels with layer thicknesses ranging between 3 m close to the surface and 12 m at 250 m depth. The latter was the
maximum depth in the model.
**2.2 Catchment models**
In BalticAPP/CLIMSEA and ECOSUPPORT, the catchment model E-HYPE (Hydrological Predictions for the
Environment, http://hypeweb.smhi.se), a process-based, high-resolution multi-basin model applied for Europe
(Hundecha et al., 2016; Donnelly et al., 2017), and a statistical hydrological model STAT (Meier et al., 2012c),
were respectively applied to calculate river runoff and nutrient inputs under changing climate but without
considering land surface changes. The statistical model calculates river runoff as precipitation minus evaporation
over the catchment area; river-borne nutrient inputs are estimated as the product of a given nutrient concentration
and the statistically derived volume flow (Gustafsson et al., 2011; Meier et al., 2012c).

In CLIMSEA, two nutrient input scenarios, defining plausible future pathways of nutrient inputs from rivers, point
sources and atmospheric deposition, i.e. the BSAP and reference (REF) scenarios (Saraiva et al., 2019a; b), are
used (Fig. 3). In BalticAPP, nutrient input scenarios follow BSAP, REF and Worst Case (WORST) scenarios
(Saraiva et al., 2019a; b; Pihlainen et al., 2020). Finally, in ECOSUPPORT, instead of WORST a business-as-
usual (BAU) scenario is applied (Gustafsson et al., 2011; Meier et al., 2011b).

In the BSAP scenario in CLIMSEA and BalticAPP, nutrient inputs linearly decrease from the actual values in
2012 (i.e., the average for 2010–2012) to the maximum allowable input in 2020 defined by the mitigation plan.
Thereafter, nutrient inputs remain constant until the end of the century. A similar temporal evolution is defined in
ECOSUPPORT but with a reference period of 1997–2003 (Gustafsson et al., 2011; their Fig. 3.1).

In the REF scenario, in CLIMSEA and BalticAPP, nutrient inputs are calculated using E-HYPE, which considers
the impact of changing river flow on nutrient inputs but neglects any changes in land use or socioeconomic
development. These inputs correspond on average to the observed mean inputs during the period 2010–2012.

The two additional, above-mentioned scenarios on future projections, BAU and WORST, are not compared
because the corresponding input assumptions differ (see Meier et al., 2018a). However, both are characterised by
population growth and intensified agricultural practices such as land cover changes and fertiliser use (HELCOM,
2007; Zandersen et al., 2019; Pihlainen et al., 2020). In this study they are discussed only for the sake of
completeness.

A comparison of the historical (1980–2005) and future (2072–2097) periods reveals that the reductions in nutrient
inputs under the BSAP scenario are smaller in ECOSUPPORT than in BalticAPP and CLIMSEA (Meier et al.,
2018a; their Fig. 3). In ECOSUPPORT and BalticAPP/CLIMSEA using the same physical-biogeochemical model
RCO-SCOBI, input changes of bioavailable phosphorus amount to −11 ktons (Model A in Meier et al., 2018a) and
−34 ktons (Model C in Meier et al., 2018a) respectively (Table 6). Corresponding input changes in bioavailable
nitrogen are −230 and −269 ktons. Table 6 also lists the calculated changes for the other two biogeochemical
models in ECOSUPPORT, BALTSEM (Model F in Meier et al., 2018a) and MOM-ERGOM (Model D in Meier
et al., 2018a), and for the REF scenarios. A comparison confirms the considerable differences between
ECOSUPPORT and BalticAPP/CLIMSEA scenario simulations. In the next section, the Baltic Sea models are
introduced.
**2.3 Baltic Sea models**
This study used data from three different Baltic Sea models. The Swedish Coastal and Ocean Biogeochemical
model coupled to the Rossby Centre Ocean model (RCO-SCOBI) is driven by the atmospheric surface field data
calculated by either RCAO or RCA4-NEMO and by the river runoff and nutrient input scenarios derived from
either STAT or E-HYPE projections and atmospheric deposition (Fig. 2). Atmospheric depositions are assumed
to be constant or reduced as in the BSAP (Fig. 3). RCO is a Bryan-Cox-Semtner-type ocean circulation model
with horizontal and vertical grid resolutions of 3.7 km and 3 m respectively (Meier et al., 1999; 2003; Meier, 2001;
2007). SCOBI is a biogeochemical module of the nutrient-phytoplankton-zooplankton-detritus (NPZD) type; it
considers state variables such as phosphate, nitrate, ammonium, oxygen concentration, the phytoplankton
concentrations of three algal types (diatoms, flagellates and others, cyanobacteria) and detritus (Eilola et al., 2009;
Almroth-Rosell et al., 2011; 2015). RCO-SCOBI has been used in many Baltic Sea climate applications (for an
overview see Meier and Saraiva, 2020), evaluated with respect to measurements and compared with other Baltic
Sea models (Eilola et al., 2011; Placke et al., 2018; Meier et al., 2018a).

The Ecological ReGional Ocean Model (ERGOM, see www.ergom.net) is a marine biogeochemical model
coupled with an ocean general circulation model and a Hibler-type sea-ice model (MOM, Griffies, 2004); its
complexity is roughly the same as that of the RCO-SCOBI model. The horizontal resolution of the model is ~5.6
km and thus somewhat coarser than that of RCO-SCOBI but in the surface layer its vertical resolution is higher,
i.e. 1.5 m in the upper 30 m and below that depth gradually increasing to as high as 5 m (Neumann et al., 2012).
The BAltic sea Long-Term large-Scale Eutrophication Model (BALTSEM) spatially resolves the Baltic Sea into
13 dynamically interconnected and horizontally averaged sub-basins with high vertical resolution (Gustafsson et
al., 2012). For further details of these and other available Baltic Sea ecosystem models the reader is referred to
Meier et al. (2018a).
**2.4 Scenario simulations**
In CLIMSEA, we analysed the ensemble of 48 RCO-SCOBI scenario simulations for the period 1976–2098 (Table
3) that was produced following the dynamical downscaling approach described in Sections 2.1–2.3 (Fig. 2) and
presented in Meier et al. (2021). Unlike in previous studies (Meier et al., 2011b; Saraiva et al., 2019a), the
CLIMSEA scenario simulations also consider various scenarios of global SLR. In the three SLR scenarios starting
from the year 2005 that were applied by Meier et al. (2021), the mean sea level changes relative to the seabed
projected by the year 2100 are: (scenario 1) 0 m, (scenario 2) the ensemble mean of RCP4.5 (0.54 m) and RCP8.5
(0.90 m) IPCC projections (IPCC, 2019b; Hieronymus and Kalén, 2020) and (scenario 3) the 95th percentiles of
the lowest case (1.26 m, here combined with RCP4.5) and highest case (2.34 m, here combined with RCP8.5)
scenarios following Bamber et al. (2019; Table 3). A deepening of the water depth at all grid points every 10 years
increases the relative sea level linearly. The spatially varying land uplift was not considered. For details, the reader
is referred to Meier et al. (2021).
The CLIMSEA ensemble simulations are compared with earlier ensemble scenario simulations by Meier et al.
(2011b; 2012c) and Neumann et al. (2012) (ECOSUPPORT), and by Saraiva et al. (2019a, b) and Meier et al.
(2019a) (BalticAPP). Both ECOSUPPORT and BalticAPP rely on a downscaling approach similar to that used in
the CLIMSEA projections (Fig. 2). However, the scenario simulations of ECOSUPPORT are based upon different
global and regional climate models, three coupled physical-biogeochemical models for the Baltic Sea and previous
GHG emission scenarios as detailed by the Fourth IPCC Assessment Report (AR4; Table 1). Compared to
BalticAPP, the CLIMSEA ensemble is enlarged by three SLR scenarios (Table 3) whereas previous projections
assumed no change in the mean sea level relative to the seabed. The inclusion of SLR scenarios followed the
finding that the relative sea level above the sills in the entrance area limits transport and controls salinity in the
entire Baltic Sea (Meier et al., 2017). As the relative SLR during the period 1915–2014 was estimated to be 0–1
mm year$^{-1}$, resulting from the net effect of past eustatic SLR and land uplift (Madsen et al., 2019), a lowest-case
scenario for the future would be a water level above the sills that is relatively unchanged (Meier et al., 2021). In
CLIMSEA, mean and highest-case scenarios follow the median values of the RCP4.5 and RCP8.5 ensembles
reported by Oppenheimer et al. (2019) and the 95[th] percentiles of the lowest- and highest-case scenarios of Bamber
et al. (2019; Table 3).
**2.5 Analysis**
*Evaluation of the historical period*
In this study, the model results of the BalticAPP and CLIMSEA scenario simulations during the historical period
were evaluated by calculating the annual and seasonal mean biases during the historical period obtained with RCO-
SCOBI simulations and reanalysis data (Liu et al., 2017). Liu et al. (2017) utilised the Ensemble Optimal
Interpolation (EnOI) method to integrate profiles of temperature, salinity and the concentrations of oxygen,
ammonium, nitrate and phosphate determined by the Swedish environmental monitoring program into the RCO-
SCOBI model. As reanalysis data are available for the period 1971–1999, we limited our bias calculations to 1976–
1999, the overlap period between the historical period of the scenario simulations and the reanalysis data. Model
data of historical periods of BalticAPP and ECOSUPPORT scenario simulations were evaluated by Saraiva et al.
(2019a, b) and Meier et al. (2011b; 2012c, d) respectively.
*Mixed-layer depth*
The mixed-layer depth (MLD) was calculated following de Boyer Montégut et al. (2004), using a threshold value
for the difference between the near-surface water temperature at 10 m depth and the temperature at the MLD of
$\Delta T = 0.2°C$.
*Secchi depth*
Secchi depth (SD) is a measure of water transparency and is calculated from $SD = 1.7/k(PAR)$, where $k(PAR)$ is
the coefficient of underwater attenuation of the photosynthetically available radiation (Kratzer et al., 2003). Factors
controlling $k(PAR)$ in the RCO-SCOBI model are the concentrations of phytoplankton and detritus. In addition,
salinity is used in one of the other Baltic Sea models (MOM-ERGOM) of the ECOSUPPORT scenario simulations
as a proxy of the spatio-temporal dynamics of coloured dissolved organic matter or yellow substances.
*Trends*
First, the monthly average of SST was computed from the model output every 48 h. The linear trend was then
calculated using the Theil-Sen estimator (Theil, 1950; Sen, 1968). The trend computed with this method was the
median of the slopes determined by all pairs of sample points. The advantage of this computationally expensive
method is that it is much less sensitive to outliers. The significance of the SST trends was evaluated from a Mann-
Kendall non-parametric test with a threshold of 95%. The SST trends were computed by season and annually. In
the latter case, the annual cycle was removed before the linear trend was computed.
Following Kniebusch et al. (2019), we performed a ranking analysis to identify the atmospheric drivers others than
air temperature that are most important for the monthly variability of SST in each ESM forcing of the CLIMSEA
data set and in the RCP scenarios RCP4.5 and RCP8.5. The SST trend is dominated by the trend in air temperature.
Thus, to eliminate the air temperature effect on SST, the difference between the SSTs and a linear regression
between the SSTs and surface air temperatures (SATs) was calculated. This was followed by applying a cross-
correlation analysis of the residual SSTs to determine the main factor driving the SST trend. For each grid point
and variable (i.e. cloudiness, latent heat flux and u-v wind components), the explained variance was calculated and
the variable explaining the most variance was identified.
*Marine heat waves*
During recent decades, the Baltic Sea region has warmed faster than either the global mean warming (Rutgersson
et al., 2015; Kniebusch et al., 2019) or any other coastal sea (Belkin, 2009), making it prone to marine heat waves
(MHWs). Indeed, short periods of abnormally high water temperatures have been documented for the Baltic Sea
(Suursaar, 2020). MHWs can be defined with reference to the mean climatology (e.g. the 90th, 95th, 98th percentile

temperature) or by temperatures exceeding absolute temperature thresholds, defined with respect to the end-user application (Hobday et al., 2018). In most cases, MHWs are defined by the number of periods, their intensity, their duration and the specific purpose (Hobday et al., 2018). In this study, the focus was on the general impact of climate change and the sensitivity of ecosystem dynamics. Hence, MHWs are defined herein as periods of SST $\geq$ 20°C lasting for at least 10 consecutive days. For comparison, we showed also MHWs defined as periods of SST exceeding the 95th percentile of the SST distribution also lasting for at least 10 consecutive days.

## 3 Results

### 3.1 Historical period

#### 3.1.1 Water temperature

The climate of the Baltic Sea region varies considerably, due to maritime and continental weather regimes. For the period 1970–1999, the annual mean SST was ~7.8°C (Fig. 4). The mean seasonal cycle of the SST is pronounced. Thus, every winter, the northern Baltic Sea, including the Bothnian Bay, Bothnian Sea and the eastern Gulf of Finland, is typically covered by sea ice (not shown). Due to its large latitudinal extension, the Baltic Sea is characterised throughout the year by a distinct SST difference between the colder northern and warmer southern sub-basins (Fig. 4). In the southern Baltic Sea, there is also a pronounced west–east temperature gradient, mainly during summer and autumn, which reflects the large-scale cyclonic circulation that transports warmer, more saline southern waters along the eastern coast and colder, less saline northern waters along the western side (see Gröger et al., 2019, their Suppl. Mat. S1; Fig. 4).

On average, during the period 1976–2005, the climate in the CLIMSEA simulations is warmer than the climate according to the reanalysis data (Fig. 4). During spring and summer, the shallow coastal zone of the northern and eastern Baltic Sea is too warm. The spatially averaged biases during winter, spring, summer and autumn and in the annual mean are 0.8, 0.9, 0.8, 1.0 and 0.9°C. The reason for the warm bias is likely a bias of the RCSM. If driven by the reanalysis data ERA-40 (Uppala et al., 2005), RCA4-NEMO systematically overestimates water temperatures and underestimates sea-ice cover in the Baltic Sea for the period 1976–2005 (Gröger et al., 2019; their Suppl. Mat. S1).

In the ECOSUPPORT scenario simulations, there is also a systematic warm bias of the RCAO driven by GCMs at the lateral boundaries, such that winter water temperatures are too warm and sea-ice cover is too low (Meier et al., 2011d, d; 2012c, d). While these biases occur in all three applied Baltic Sea models (Table 3) forced with RCSM atmospheric surface fields, in the simulations driven by regionalised reanalysis data (ERA-40) the mean biases are smaller (Eilola et al., 2011).

#### 3.1.2 Mixed-layer depth

Figure 5 shows the seasonal MLD cycle calculated after de Boyer Montégut et al. (2004). A deeper MLD with pronounced west-east gradients characterises the open ocean. This is related to the predominant southwesterly wind regime, with the larger wind fetches and higher significant wave heights in the eastern Gotland Basin causing

wave-induced vertical mixing. Furthermore, a positive sea-atmosphere temperature contrast favours higher wind
speeds ('positive winter thermal feedback loop'; Gröger et al., 2015; 2021b). In spring, a weakening wind regime,
which reduces heat exchange (with a shift from heat loss to heat gain), together with the increased solar irradiance
leads to a thinner MLD in the southern Baltic Sea while melting sea ice and subsequent thermal convection and
wind-induced mixing maintain a MLD > 50 m in the sea's northern part. During summer, when atmosphere-ocean
dynamics are weakest, a pronounced thermocline develops and MLDs are shallowest (the 'summer thermal short
circuit'; Gröger et al., 2021b). During autumn, the atmosphere cools faster than the Earth's surface, and land
masses faster than open sea areas. These increased thermal differences result in a stronger large-scale wind regime
with a positive feedback on the MLD.

The ensemble model mean in CLIMSEA reproduces these dynamics and the spatial pattern relatively well. During
the cold season, however, the MLD is somewhat shallower in the simulation than in the reanalysis data of Liu et
al. (2017). This may be the result of air-sea coupling. Gröger et al. (2015, 2021b) demonstrated that the complex
thermal air-sea feedbacks in winter are less well represented by stand-alone ocean models than by fully coupled
ocean-atmosphere GCMs. This can result in SST biases and too-shallow MLDs (Gröger et al., 2015; Fig. 7a
therein; Gröger et al., 2021b). However, the real reasons for the underestimated winter MLD are unknown.

In the literature, MLDs in the ECOSUPPORT scenario simulations have not been analysed.
**3.1.3 Marine heat waves**
Baltic Sea MHWs are defined herein as periods of >10 days duration during which 1) the SST is > 20°C and 2)
the SST exceeds the 95th percentile temperature. The CLIMSEA climate model ensemble mean and the reanalysis
data set generated by the same model are compared in Figure 6 (Liu et al., 2017).

The first MHW index uses a fixed threshold that emphasises the environmental impact of the heat waves. In
particular, diazotrophic nitrogen fixation becomes effective at higher temperatures. The spatial pattern of MHWs
is strongly related to the simulated SST. Figure 6a shows that MHWs are mostly absent in the open sea of the
Baltic proper and further north in the Gulf of Bothnia, but they are highly abundant in shallow marginal bays such
as the Gulf of Finland and Gulf of Riga as well as along the coasts. The MHWs produced by the RCO ensemble
mean are generally more frequent and of longer duration than those of the reanalysis data set. Furthermore, the
coastal signature of high abundance extends further offshore (Fig. 6a). For the Belt Sea and Bay of Lübeck, this
leads to considerable deviations from the reanalysis data set.

The second index is based on a reference climatology, here defined as that of 1976–1999. The number of MHWs
(Fig. 6c) correlates negatively with their average duration (Fig. 6d). This is somewhat more pronounced in the
reanalysis data set. In general, the patterns obtained with the reanalysis data and the RCO are similar but the
amplitude of spatial variance is higher in the former (Fig. 6c), as it includes small-scale regional observations. In
the RCO (Fig. 6d), MHWs in the open sea are of the longest duration, with their interruption likely due to the
vertical mixing induced by wind events.

Since MHWs in the Baltic Sea are predominantly a summer phenomenon, the stability of the seasonal thermocline
is likely a key element in their dynamics such that processes related to vertical mixing can be considered a
benchmark in their simulation by the models. Given that mixing is highly parameterised in current ocean models,
the RCO reproduces the spatial patterns of the number and average duration of MHW reasonably well.
In the literature, MHWs in the ECOSUPPORT scenario simulations have not been analysed.
**3.1.4 Salinity**
The annual mean sea surface salinity (SSS) distribution shows a large north–south gradient mirroring both the
input of freshwater from rivers, mostly located in the northern catchment area, and saltwater inflows from the
North Sea (Fig. 7). The SSS drops from about 20 g kg$^{-1}$ in the Kattegat to < 2 g kg$^{-1}$ in the northern Bothnian Bay
and eastern Gulf of Finland. For the period 1970–1999, the annual mean SSS of the Baltic Sea including the
Kattegat was ~7.3 g kg$^{-1}$. Large inflows of heavy saltwater from the Kattegat occasionally ventilate the bottom
water of the Baltic Sea, filling its deeper regions (Fig. 7). As tides are almost absent, mixing is limited such that
the water column is characterised by a pronounced vertical gradient in salinity, and consequently also in density,
between the sea surface and the bottom.
Probably due to differences in the data of the hydrological model (E-HYPE) compared to observations, SSS in the
coastal zone and the Kattegat is on average lower in the CLIMSEA climate models than in the reanalysis data of
Liu et al. (2017) (Fig. 7). The spatially averaged, annual mean bias is −0.4 g kg$^{-1}$. Bottom salinities in the Belt
Sea, Great Belt area and the Gotland Basin (especially in the northwestern part) are considerably higher and in the
Bornholm Basin considerably lower in the climate models than in the reanalysis data (Fig. 7). The spatially
averaged, annual mean bias is +0.3 g kg$^{-1}$. Hence, vertical stratification in the Belt Sea, Great Belt area and the
Gotland Basin is also larger in the climate models than in the reanalysis data, because the difference between
surface and bottom salinities is a good proxy for vertical stratification.
In the ECOSUPPORT scenario simulations, SSS is overestimated in the entire Baltic Sea, in particular in its
northern and eastern regions (Meier et al., 2011c; 2012c). In both, the ensemble mean bottom salinity and vertical
stratification are also overestimated while the bottom salinity in the eastern Gotland Basin is well reproduced
(Meier et al., 2012c).
**3.1.5 Sea level**
Due to the seasonal cycle in wind speed, with wind directions predominantly from the southwest, the sea level in
the Baltic Sea varies considerably throughout the year, with the highest levels (~40 cm), measured relative to the
Kattegat, occurring during winter at the northern coasts of the Bothnian Bay and at the eastern coasts of the Gulf
of Finland (Fig. 7). For the period 1976–1999, the annual mean sea level in the Nordic height system 1960 (NH60)
as determined by Ekman and Mäkinen (1996) was ~16 cm, with a horizontal north–south difference of ~35 cm
(not shown). This sea level slope was explained by the lighter brackish water in the northeastern Baltic Sea than
in the Kattegat and by wind coming from the southwesterly direction, which pushes the water to the north and east
(Meier et al., 2004a).

The differences in the mean sea level between the CLIMSEA climate models and the reanalysis data are small
(Fig. 7) and the spatially averaged, winter mean bias is only +0.6 cm. Sea levels in some parts of the coastal zone
such as the western Bothnian Sea are higher in the climate models than in the reanalysis data, probably due to
lower salinities. The negative sea level bias in the eastern Gotland Basin suggests an intensified, basin-wide
cyclonic gyre. The seasonal cycle of the ensemble mean sea level is relatively well simulated, but with an
overestimated sea level in early spring and an underestimated sea level in summer at all investigated tide gauge
locations compared to observations and to a hindcast simulation driven by regionalised ERA40 data (Fig. 8).

In the ECOSUPPORT scenario simulations, sea levels were not systematically analysed. In one of the three models
(RCO-SCOBI), seasonal mean biases were comparable to the biases in the CLIMSEA scenario simulations (Meier
et al., 2011a).
**3.1.6 Oxygen concentration and hypoxic area**
Since the 1950s, nutrient inputs into the Baltic Sea have increased due to population growth and intensified
fertiliser use in agriculture (Gustafsson et al., 2012; Fig. 3). Nutrient inputs reached their peak in the 1980s but
have steadily declined following the implementation of nutrient input abatement strategies. Nonetheless, since the
1960s, the bottom water of the Baltic Sea below the permanent halocline has been characterised by oxygen
depletion and large-scale hypoxia (Fig. 9).

Consistent with the stratification biases in the deeper sub-basins of the Baltic Sea, summer bottom oxygen
concentrations in the Bornholm Basin are higher and those in the Gotland Basin lower in the CLIMSEA/BalticAPP
climate simulations than in the reanalysis data of Liu et al. (2017) (Fig. 9). The stronger vertical stratification,
especially at the halocline depth, hampers vertical fluxes of oxygen, causing prolonged residence times and lower
bottom oxygen concentrations. Spatially averaged biases during winter, spring, summer and autumn and in the
annual mean are small but systematic: −0.6, −0.7, −0.7, −0.5 and −0.6 mL L$^{-1}$ respectively.

In the ECOSUPPORT scenarios, the ensemble mean deep-water oxygen concentration in the eastern Gotland
Basin is slightly higher (but within the range of natural variability) and that in the Gulf of Finland significantly
lower than determined from observations (Meier et al., 2011b; 2012d).
**3.1.7 Nutrient concentrations**
Nutrient (i.e. phosphorus and nitrogen) content in the surface layer during winter is a good indicator of the intensity
of the following spring bloom. Sea-surface mean winter concentrations of phosphate and nitrate are highest in the
coastal zone, in particular close to the mouths of the large rivers in the southern Baltic Sea that transport elevated
inputs of nutrients into the sea (Fig. 9).

For the historical period of 1976–1999, winter surface phosphate concentrations according to the climate
simulations are relatively close to those of the reanalysis data (Fig. 9). The concentrations differ substantially only
in those coastal regions influenced by large rivers, such as those affected by discharges of the Odra, Vistula and
Pärnu rivers. Spatially averaged biases are largest during summer and autumn, with an average bias in summer of
+0.2 mmol P m$^{-3}$.

Likewise, winter surface nitrate concentrations in the simulations are close to those in the reanalysis data but in
coastal regions they differ due to differences in the inputs from large rivers (Fig. 9). This is exemplified by the
Gulf of Riga and the eastern Gulf of Finland, where the large differences between them are due to inputs from the
Neva River. Spatially averaged biases during winter, spring, summer, autumn and in the annual mean are rather
small but systematic: −1.1, −1.3, −0.5, −0.7 and −0.9 mmol N m$^{-3}$ respectively.

In the ECOSUPPORT scenario simulations, the simulated profiles of phosphate, nitrate and ammonium are within
the range of observations for 1978–2007, except in the case of phosphate in the Gulf of Finland (Meier et al.,
2012d). According to hindcast simulations, the biases in the coupled physical-biogeochemical models of the Baltic
Sea relative to the standard deviations of observations are larger for the northern Baltic Sea than for the Baltic
proper (Eilola et al., 2011).
**3.1.8 Phytoplankton concentrations**
During the period 1976–1999, dense phytoplankton blooms were confined to the coastal zone, i.e. the area with
the highest nutrient concentrations (Fig. 10). Water transparency, measured by Secchi depth, is lower in the Baltic
Sea than in the open ocean (Fleming-Lehtinen and Laamanen, 2012), and for the period 1970–1999 the annual
mean Secchi depth averaged for the entire Baltic Sea, including the Kattegat, was only ~6.6 m. The Secchi depth
is also much smaller in the coastal zone than in the open Baltic Sea (Fig. 10), and in the northern Baltic Sea than
in the Gotland Basin, attributable to yellow substances originating from land (Fleming-Lehtinen and Laamanen,

484     2012).


Due to nutrient concentration biases, the annual mean surface phytoplankton concentrations of the simulations are
close to those of the reanalysis data of Liu et al. (2017) but they deviate in coastal regions (Fig. 10). Spatially
averaged biases during winter, spring, summer, and autumn and in the annual mean are relatively small: +0.02,
−0.1, −0.009, +0.06 and −0.008 mg chlorophyll (Chl) m$^{-3}$ respectively. Note that the reanalysis data of Liu et al.
(2017) assimilate nutrient and oxygen concentrations but not chlorophyll data.

Similar results are found for the mean biases in the simulated Secchi depths (Fig. 10). In climate simulations,
Secchi depths are systematically deeper in the regions south of Gotland and at the entrance to the Gulf of Finland
(northeastern Gotland Basin) than elsewhere in the Baltic Sea. Spatially averaged biases during winter, spring,
summer and autumn and in the annual mean are +0.2, +0.4, +0.06, +0.1 and +0.2 m respectively.

Compared to the Secchi depth data from HELCOM (HELCOM, 2013; their Table 4.3) and Savchuk et al. (2006;
their Table 3), the CLIMSEA climate simulations under- and overestimate the Secchi depth in the southwestern
and northern Baltic Sea respectively, while in the Gotland Basin the model results well fit the observations (Meier
et al., 2019a).

In the ECOSUPPORT scenario simulations, Secchi depth was not compared with observations.
**3.1.9 Biogeochemical fluxes**
An evaluation of biogeochemical fluxes, such as primary production and nitrogen fixation, is difficult because
observations are lacking. An exception is the study by Hieronymus et al. (2021), in which historical simulations
with RCO-SCOBI were compared with in situ observations of nitrogen fixation. The RCO-SCOBI model includes
a cyanobacteria life cycle (CLC) model (Hense and Beckmann, 2006; 2010) driven by reconstructed atmospheric
and hydrological data. The authors found a satisfactory agreement, with the results mainly within the uncertainty
range of the observations. However, simulated monthly mean nitrogen fixation during 1999–2008 showed a
prolonged peak period in July and August whereas according to observations the peak was mostly confined to
July. It should be noted that the RCO-SCOBI version used in the scenario simulations discussed here (e.g., Saraiva
et al., 2019a) does not contain a CLC model.
**3.2 Future period**
**3.2.1 Water temperature**
*Annual and seasonal mean changes*
In Figures 11 and 12 and Table 7, annual and seasonal mean SST changes between 1976–2005 and 2069–2098 in
RCO-SCOBI are depicted and quantified respectively. The maximum seasonal warming signal propagates
between winter and summer from the Gulf of Finland via the Bothnian Sea into the Bothnian Bay (Fig. 11).
Maximum warming occurs during summer in the Bothnian Sea and Bothnian Bay. The seasonal patterns of RCP4.5
and RCP8.5 are similar although warming is greater in the latter. As SLR has almost no impact on SST changes,
BalticAPP and CLIMSEA scenario simulations yield similar results (not shown). The warming level according to
ECOSUPPORT is between that predicted by CLIMSEA/BalticAPP RCP4.5 and RCP8.5 because the GHG
emissions of the A1B scenario, which forces the ECOSUPPORT ensemble[2], are between those of the RCP4.5 and
RCP8.5 scenarios.

In the CLIMSEA/BalticAPP RCSM projections, the annual mean SST changes in the Baltic Sea driven by four
ESMs, i.e. MPI-ESM-LR, EC-EARTH, IPSL-CM5A-MR, HadGEM2-ES, under the RCP8.5 scenario are +2.3,
+3.7, +3.5 and +4.7°C respectively (Gröger et al., 2019). Thus, the ensemble mean change is +3.5°C. The
corresponding ensemble mean change in the RCO-SCOBI scenario simulations is smaller, +2.9°C. Different
MLDs, vertical stratification and sea-ice cover in the two ocean models, RCO-SCOBI and NEMO, may explain
the different responses. Indeed, a comparison of the MLD between the two models reveals a shallower MLD in
the RCSM than in RCO-SCOBI (not shown), which argues for a higher sensitivity of the RCSM to climate
warming.

---

[2]One of the scenario simulations of ECOSUPPORT is driven by the A2 scenario, which due to higher GHG emissions is generally warmer than the A1B scenario. However, this particular simulation of the ECHAM5–MPIOM GCM is exceptional and at the end of the 21st century the temperature is not much warmer than that obtained with the corresponding run based on the same model under the A1B scenario.

While the spatial patterns of the SST changes in the scenario simulations of ECOSUPPORT (e.g., Meier et al.,
2012c) and CLIMSEA (e.g., Saraiva et al., 2019b) are similar, the uncertainties due to the applied global (Meier
et al., 2011a) or regional (Meier et al., 2012b) model are in some cases considerable. Of note is the summer
ensemble range of the various GCMs (Meier et al., 2011a). The strong dependence on forcing is seen by comparing
the different warming levels in the RCP4.5 and RCP8.5 scenarios shown in Figure 12.

*Trends*
Since SLR and nutrient input scenarios have a negligible impact on SST changes, only the RCP4.5 and RCP8.5
scenarios in CLIMSEA/BalticAPP are compared. The multi-model mean of the annual mean SST trends averaged
over the Baltic Sea is ~0.18 °C decade$^{-1}$ and ~0.35 °C decade$^{-1}$ in the RCP4.5 and RCP8.5 scenarios respectively
(Fig. 13a, f). At the Baltic Sea scale, seasonal SST trends based on annual values vary only slightly (±0.01 °C
decade$^{-1}$ in both scenarios). However at the sub-basin scale, seasonal variations are much stronger, reaching ±0.05
°C decade$^{-1}$ in the northern Baltic Sea, with a maximum in summer (Fig. 13). This summer maximum can be
explained by the projected decline in sea-ice cover in summer, as occurred during the period 1850–2008
(Kniebusch et al., 2019).

As seen in Figure 14, the relative SST trends indicate faster warming of the northern than the southern Baltic Sea
(0.02 °C decade$^{-1}$ and 0.04 °C decade$^{-1}$ in the RCP4.5 and RCP8.5 scenarios respectively), with the largest trends,
calculated over the entire period 2006–2099, reaching ~0.24 °C decade$^{-1}$ and ~0.45 °C decade$^{-1}$ in RCP4.5 and
RCP8.5 respectively. However, a calculation of the SST trends by 30-year slice periods every 10 years over the
entire period shows that annual SST trends are variable over time (not shown). The natural variability appears to
modulate these trends, with successive periods of increasing and decreasing SST trends over a period of about 30
years. For example, in the RCP8.5 scenario, SST trends gradually increase over the first 50 years of the period,
reaching a maximum of 0.5 °C decade$^{-1}$ between 2046 and 2075, before declining slightly from 2060 onwards. As
in the RCP4.5 scenario, this is a result of the pronounced natural variability in this scenario. Despite the robustness
of the spatial pattern of the SST trends ($p < 0.05$ everywhere), an analysis of SST trends for the four ESM forcings
reveals an important dependency of those trends on atmospheric forcings, with a spread of ±0.06 °C decade$^{-1}$ from
the multi-model mean of both scenarios (not shown).

At an annual timescale, the variability in the air temperature, through the sensible heat fluxes, is the main driver
of the Baltic Sea's SST (Kniebusch et al., 2019), illustrated here by the high variance of SST explained by air
temperature (between 0.85 and 0.95, Fig. 15). The minimum of variance explained is located in the Bothnian Bay,
where the sea-ice cover isolates seawater from the air in winter.

The processes responsible for the SST trends were analysed using a rank analysis of atmospheric variables (i.e.
latent heat fluxes, cloud cover and u-v wind components) following Kniebusch et al. (2019; Fig. 16). The second
parameter (after SAT) explaining the variability in the SST differs according to the location and ESM.
Nevertheless, in all ESMs and in both RCP scenarios, zonal and meridional wind components are the variables
that best correlate with SST along most of the coastal areas, probably because of upwelling. In the open sea of the
Baltic proper and in the Bothnian Bay, the second most important variable is cloudiness. This is also the case in
the Bothnian Sea under the RCP4.5 scenario. However, in RCP8.5 the second most important variable at this
location is the latent heat flux. The difference is perhaps due to the absence of sea ice, and therefore the amplified
air-sea exchange, under RCP8.5.

In the vertical, temperature trends are larger in the surface layer than in the winter water of the Baltic Sea above
the halocline, thus causing a more intense seasonal thermocline (see Section 3.2.2). Surface layer trends are largest
in spring and summer (not shown). Elevated trends also characterise deep water, due to the influence of saltwater
inflows that will be warmer in a future climate because they originate from the shallow entrance area and occur
mainly in winter. Hence, in sub-basins that are sporadically ventilated by lateral saltwater inflows, such as the
Bornholm Basin and the Gotland Basin, the deep water below the halocline will warm more than the overlaying
intermediate layer water.

In the literature, trends in ECOSUPPORT scenario simulations have not been analysed.
**3.2.2 Mixed-layer depth**
Figure 17 shows the changes in the MLD. During winter, reduced sea-ice cover in the Bothnian Sea and Bothnian
Bay favours a widespread deepening of their MLDs, likely caused by wind-induced mixing. In spring, the most
pronounced feature is a strong shallowing of the MLD in the Bothnian Sea, probably attributable to the radiative
fluxes that warm the surface layer and to less thermal convection (Hordoir and Meier, 2012). During the historical
period, water temperatures in this area were between 2.0 and 3.0°C (Fig. 4). Thus, in the future, surface water
warming between 1.6 and 2.4°C (Fig. 11) may hamper thermal convective mixing.

The changes during summer are less pronounced. In contrast to winter, there is an overall shallowing in the entire
Baltic Sea. This is in agreement with a shallower, more intense thermocline in warming scenarios, as suggested
by Gröger et al. (2019), and it is a common feature among the projections, because the changes in wind speed are
small (Christensen et al., 2021). Autumn is primarily characterised by a prolongation of the thermal stratification,
leading to an overall shallower MLD than during the historical period.

While Hordoir et al. (2018; 2019) speculated that these changes in thermocline depth during summer will impact
the vertical overturning circulation, the meridional overturning circulation in the Baltic proper does not show a
clear signal but rather a northward expansion of the main overturning cell (Gröger et al., 2019). Indeed, the effect
is expected to be small (Placke et al., 2021).
**3.2.3 Marine heat waves**
The number of MHWs within climatological 30-year time slices is shown in Figure 18. Under historical climate
conditions, MHWs are virtually absent in open ocean areas. They are most frequent in shallow regions and more
abundant along the eastern (Baltic States) than the western (Swedish) coasts, which may reflect the greater
frequency of coastal upwelling events along the western than the eastern coasts of the Baltic Sea. Even under the
RCP4.5 scenario, wide areas of the Baltic proper are affected by MHWs roughly once a year. The strongest
response is projected for the high-emission RCP8.5 scenario, and specifically in marginal basins such as the Gulf
of Riga and the Gulf of Finland, where in the future MHWs will occur two or three times per year. Not only the

frequency but also the average duration of the MHWs will increase with climate warming. Under RCP8.5, MHWs of ~20 days duration will occur even in the open Gulf of Bothnia (Fig. 18). This increase in MHWs is likewise linked to an increased frequency of tropical nights in the Baltic Sea (Meier et al., 2019a; Gröger et al., 2021b).

MHWs can also be analysed by calculating them with respect to the 95th percentile temperature of the historical reference climate (Fig. 19). For the historical climate, the average duration of MHWs in most regions is < 20–30 days, although in the southern Baltic Sea, especially west of the Baltic proper, MHWs are more frequent. However, the climate change signal is characterised by MHWs that are both more frequent and of longer duration. In RCP4.5, MHWs in the Baltic Sea occur at least every year. The strongest increase in frequency is near the coasts, but the average duration increases less than in the open sea (Fig. 19). This is probably related to repeated cold-water entrainments from the open sea that interrupt warm periods because of the larger variability in the coastal zone than in the open sea. In addition, with their lower heat storage capacity, shallow areas are more sensitive to cold weather events and the associated oceanic heat loss.

### 3.2.4 Salinity

In the CLIMSEA ensemble, salinity changes are not robust, i.e. the ensemble spread is larger than the signal (Meier et al., 2021). Under both RCP4.5 and RCP8.5, the ensemble mean salinity change is small because the impact on salinity of the projected increase in total river runoff from the entire catchment (Fig. 3) is approximately compensated by the impact of larger saltwater inflows due to the projected SLR (Table 8). Hence, compared to previous studies such as those by Meier et al. (2011b; ECOSUPPORT) and Saraiva et al. (2019a; BalticAPP; Fig. 12), the ensemble mean salinity changes in CLIMSEA are much smaller (Table 8). In idealised sensitivity experiments performed with the RCO-SCOBI model for the period 1850–2008 (Meier et al., 2017; 2019d), the change in the average Baltic Sea salinity (1988–2007) increased linearly with SLR and at a rate of ~1.4 g kg$^{-1}$ m$^{-1}$ (Table 9).

### 3.2.5 Sea level

Following global sea level changes, SLR in the Baltic Sea will accelerate (Hünicke et al., 2015; Church et al., 2013; Bamber et al., 2019; Oppenheimer et al., 2019; Weisse and Hünicke, 2019), albeit at a somewhat slower rate than the global mean because of the remote impact of the melting Antarctic ice sheet (Grinsted, 2015). Changes in SLR in the North Atlantic (and the Baltic Sea) will be larger in response to the melting of the Antarctic ice sheet than to the melting of Greenland, due to gravitational effects. For a mid-range scenario, SLR in the Baltic Sea is projected to be ~87% of the global mean (Pellikka et al., 2020). Land uplift will partly, or even more than fully, compensate for the eustatic SLR, in particular in the northern Baltic Sea (e.g. Hill et al., 2010). In RCP2.6 and RCP8.5, the global mean sea level in 2100 is 43 cm and 84 cm higher than during the period 1986–2005 (Oppenheimer et al., 2019). For these two scenarios, likely ranges are 29–59 cm and 61–110 cm respectively. Bamber et al. (2019) assessed ice sheet dynamics in detail and subsequently estimated global-median SLRs in 2100 of 69 cm and 111 cm for low- and high-case scenarios respectively. Likely ranges according to the authors were 49–98 cm and 79–174 cm and very likely ranges 36–126 cm and 62–238 cm.

In BalticAPP and CLIMSEA scenario simulations, sea level changes are small (Fig. 12, Table 8) whereas in
ECOSUPPORT scenario simulations they are larger, particularly in spring, because one member of the multi-
model ensemble considers Archimedes' principle (not shown). Note that the sea level changes shown in Figure 12
consider only changing river runoff, changing wind, and melting sea ice as affecting the sea level according to
Archimedes' principle (only in the ECOSUPPORT ensemble); as neither the global mean SLR nor land uplift is
included, they have to be added (e.g. Meier, 2006; Meier et al., 2004a).
In CLIMSEA, there are no statistically significant seasonal changes in the SLR (Fig. 20). In both GHG
concentration scenarios, the largest changes are only about ±5 cm. According to these results, systematic changes
in the regional wind field (Christensen et al., 2021) and nonlinear effects are negligible. Instead, in the projections,
the mean absolute sea level in the Baltic Sea simply follows the mean sea level in the North Atlantic. However,
the spatially inhomogeneous isostatic adjustment will considerably alter patterns of sea level changes relative to
the sea floor.
In response to the global mean SLR, the sea level extremes in the Baltic Sea that are rare today will become more
common in the future (e.g. Hieronymus and Kalén, 2020). However, changes in sea level extremes relative to the
mean sea level will not be statistically significant because wind velocities are projected to remain unchanged
(Christensen et al., 2021). The exceptions are areas with sea-ice decline since they are linked to a decrease in
atmospheric stability accompanied by increased wind velocities, the result of increases in temperature and
turbulent fluxes (Meier et al., 2011c). These increases will mostly translate as changes from calm to light wind
conditions as the stable atmospheric boundary layer becomes less stable. For stronger wind conditions related to
high sea level extremes, the impact of stratification effects on mixing is small. In addition, open water areas after
sea-ice loss have a smaller surface roughness than ice-covered areas, with the reduced surface friction leading to
an increase in wind velocities.
As sea level extremes also depend on the path of low-pressure systems over the Baltic Sea area (Lehmann et al.,
2011; Suursaar and Sooäär, 2007), which in a future climate do not show systematic changes (Christensen et al.,
2021), changes in sea level extremes relative to the mean sea level are not expected. In addition, a large internal
variability at low frequencies prevents the detection of climate-warming-related changes in sea level extremes
(Lang and Mikolajewicz, 2019).
**3.2.6 Oxygen concentration and hypoxic area**
*Bottom oxygen concentration*
Projected changes in bottom oxygen concentrations differ considerably between ECOSUPPORT and
BalticAPP/CLIMSEA scenario simulations, as illustrated for summer (Figs. 21 and 22, Table 10), whereas the
differences between BalticAPP (SLR = 0 cm) and CLIMSEA (SLR > 0 cm) scenarios are smaller (Meier et al.,
2021). The differences between the ECOSUPPORT and BalticAPP ensembles mainly reflect the different
experimental setups of the simulations and the different nutrient input scenarios (Meier et al., 2018a). While in
shallow regions without a pronounced halocline future bottom oxygen concentrations decrease in all scenario
simulations, due to the lower oxygen saturation concentrations, in deeper offshore regions with a halocline,
changes in bottom oxygen concentration depend largely on the nutrient input scenario (Figs. 21 and 22). In
ECOSUPPORT scenario simulations, the future bottom oxygen concentration decreases significantly in all
scenarios except the BSAP, where in deeper regions it changes only slightly on average (see Meier et al., 2011b).
By contrast, in the BalticAPP projections, under the BSAP bottom oxygen concentrations in deeper regions
increase considerably, regardless of the degree of warming (see Saraiva et al., 2019a; Meier et al., 2011b). Under
RCP4.5, bottom oxygen concentrations increase even under the nutrient inputs of REF and WORST whereas
RCP8.5 predicts slight reductions in the Bothnian Sea and southwestern Baltic Sea, in particular under WORST.
Similar results were calculated for the CLIMSEA ensemble (Meier et al., 2021).
Most of the differences in the oxygen concentration changes between the ECOSUPPORT and
BalticAPP/CLIMSEA ensembles can be explained as follows. In ECOSUPPORT, changes in nutrient input
relative to the historical period 1961–2006, including the observed nutrient inputs averaged from the period 1995–
2002, were applied (Gustafsson et al., 2011; Meier et al., 2011b). For the historical period 1980–2002, these inputs
were lower than in BalticAPP/CLIMSEA scenario simulations because in the latter the observed monthly nutrient
inputs, including the pronounced decline from the peak in the 1980s until the much lower recent values, were used
as the forcing (Meier et al., 2018a). Furthermore, in ECOSUPPORT, future nutrient inputs under the BSAP
scenario were calculated as relative changes, resulting in higher future inputs than in BalticAPP/CLIMSEA, in
which absolute values of the BSAP were applied.
Hence, the reductions between future and historical nutrient inputs are smaller in ECOSUPPORT under the BSAP
than in BalticAPP/CLIMSEA (Table 6) and result in a smaller response of biogeochemical cycling. We argue that
the more realistic historical simulation, including a spin-up since 1850, based on observed or reconstructed nutrient
inputs as used in the BalticAPP and CLIMSEA ensembles result in a model response that is more realistic than
that of the ECOSUPPORT scenario simulations.
*Hypoxic area*
In ECOSUPPORT, the hypoxic area is projected to increase under REF and BAU nutrient input scenarios (Meier
et al., 2011b). Only under BSAP is there a slight decrease compared to the early 2000s.
In CLIMSEA under REF, the hypoxic area is projected to decrease slightly until about 2050, as a delayed response
to nutrient input reductions, and then increase again towards the end of the century, presumably in response to
increased nutrient inputs and warming (Fig. 23). Larger hypoxic areas are calculated under RCP8.5 than under
RCP4.5. Under BSAP, the hypoxic area is projected to considerably decrease. At the end of the century, the size
of the hypoxic area is expected to be 22–78% smaller than the average size during the period 1976–2005. This
range represents the results of the various scenario simulations.
In accordance with previous studies, such as Saraiva et al. (2019b) and Meier et al. (2021), the impact of warming
(reduced oxygen solubility, increased internal nutrient cycling, increased riverine inputs) and of increasing
stratification (decreased ventilation) will be an amplified depletion of oxygen that enlarges the hypoxia area in the
Baltic Sea and partially counteracts nutrient input abatement strategies such as the BSAP. However, in all available
scenarios the impact of climate change is smaller than the impact of nutrient input changes.

### 3.2.7 Nutrient concentrations

While in ECOSUPPORT scenario simulations of future climate the projected surface phosphate concentrations in winter increase under all three nutrient input scenarios (except in the Gulf of Finland in BSAP), in BalticAPP projections the surface phosphate concentrations in winter decrease almost everywhere (except in the Odra Bight and adjacent areas in REF and WORST) (not shown). In contrast to the nearly ubiquitous changes in the surface phosphate concentration, larger nitrate concentration changes are usually confined to the coastal zone and differ in their signs. In ECOSUPPORT projections, the increases in winter surface nitrate concentrations in REF and BAU are largest in the Gulf of Riga, the eastern Gulf of Finland and along the eastern coasts of the Baltic proper (not shown). In BalticAPP projections, the increases in winter surface nitrate concentrations in REF and WORST are largest in the Bothnian Bay and the Odra Bight while in the Gulf of Riga and the Vistula lagoon nitrate concentrations decrease. Overall, the differences in surface nutrient concentrations between the two sets of scenario simulations are considerable (not shown) and can be explained by the large differences in nutrient inputs from land. Thus, while the projected changes in inputs in ECOSUPPORT refer to the average inputs during 1995–2002, in BalticAPP scenario simulations the observed historical changes include a decline in nutrient inputs since the 1980s (Meier et al., 2018a).

### 3.2.8 Phytoplankton concentrations

Annual mean changes in surface phytoplankton concentration (expressed as chlorophyll concentration) follow the changes in nutrient concentrations and are confined to the productive zone along the coasts (Fig. 24). In ECOSUPPORT projections, annual mean Secchi depths decrease in all scenario simulations (see Fig. 25 and Table 11). In the BalticAPP projections, the area-averaged Secchi depths generally increase, except in the combined RCP8.5 and BAU scenario (Table 11), indicating a general improvement of the water quality in future compared to the present climate. The most striking changes occur in the BSAP scenario, in which the Secchi depth increases by up to 2 m in the coastal zone of the eastern Baltic proper. Changes in stratification (illustrated by the differences between BalticAPP and CLIMSEA ensembles and between the CLIMSEA ensemble mean and high SLR scenarios) have only a minor impact on water transparency (Table 11). The overwhelming driver of the changes in the Secchi depth are nutrient input scenarios (illustrated by the differences between ECOSUPPORT and BalticAPP/CLIMSEA ensembles and highlighted by, in some cases, contradictory signs in the changes).

### 3.2.9 Biogeochemical fluxes

In CLIMSEA under the BSAP, primary production and nitrogen fixation are projected to considerably decrease in a future climate (Fig. 23). According to this scenario, the interannual variability declines. Under REF, nitrogen fixation is projected to slightly decrease until ~2050, as a delayed response to nutrient input reductions, and then to increase towards the end of the century, likely in response to increased nutrient inputs and warming. At the end of the century, primary production and nitrogen fixation will be at the same level as under current conditions. The impact of warming is larger under high than under low nutrient conditions (Saraiva et al., 2019b).

**3.2.10 Relation to large-scale atmospheric circulation**

The dominant large-scale atmospheric pattern controlling the climate in the Baltic Sea region during winter is the North Atlantic Oscillation (NAO; Hurrell, 1995). However, its influence is not stationary but depends on other modes of variability, such as the Atlantic Multidecadal Oscillation (AMO; Börgel et al., 2020). During the past climate, the relationship between the NAO index and regional climate variables in the Baltic Sea region, such as SST, changed over time (Vihma and Haapala, 2009; Omstedt and Chen, 2001; Hünicke and Zorita, 2006; Chen and Hellström, 1999; Meier and Kauker, 2002; Beranová and Huth, 2008).

Figure 26 shows the calculated ensemble mean winter (December–February) NAO index for the period 2006– 2100. For the RCP4.5 emission scenario, the NAO shows high interannual variability. Following a wavelet analysis, the calculated NAO index exhibits decadal variability, which differs for every model (not shown). A comparison of RCP4.5 with the high-emission scenario RCP8.5 shows that the spread of the ensemble increases with increasing GHG concentrations. Figure 26 also depicts the running correlation between the NAO index and the area-averaged SST. The correlation remains positive but it is not constant in time. Also evident from a comparison of RCP4.5 and RCP8.5 is that there are no systematic changes in the two emission scenarios, although for RCP8.5 the ensemble spread is slightly larger.

**4 Knowledge gaps**

In the largest set of scenario simulations of this study, the CLIMSEA ensemble, only four ESMs were regionalised using only one RCSM; consequently, this ensemble is still too small to estimate the uncertainties caused by ESM and RCSM differences. While nine ESMs with the same RCSM were recently regionalised, they did not include running modules for terrestrial and marine biogeochemistry Gröger et al., 2021b), such that these simulations were not considered in our assessment. The uncertainties related to unresolved physical and biogeochemical processes in the Baltic Sea and on land were also not considered, because only one Baltic Sea and one catchment model were used. Although the CLIMSEA ensemble is larger than the ensembles in previous studies, it is still too small to estimate all sources of uncertainty.

In addition to the uncertainties related to global and regional climate and impact models, pathways of GHG and nutrient emissions are thus far unknown and the role of natural variability versus anthropogenic forcing is not well understood (Meier et al., 2018a; 2019b; 2021). Recent studies suggest that the impact of natural variability, such as the low-frequency AMO, is larger than hitherto estimated. For instance, in paleoclimate simulations the AMO affected Baltic Sea salinity at time scales of 60–180 years (Börgel et al., 2018), which is longer than the simulation periods of available scenario simulations. Furthermore, the AMO may also influence the centres of action of the NAO (Börgel et al., 2020). Lateral tilting of the positions of the Icelandic Low and Azores High explains the changes in the correlations between the NAO and regional variables such as water temperature, sea-ice cover and river runoff in the Baltic Sea region (Börgel et al., 2020). Despite indications that the AMO is affected by climate states such as the Medieval Climate Anomaly and Little Ice Age (Wang et al., 2017; Börgel et al., 2018), how future warming would affect these modes of climate variability is unclear.

Changes in sea-ice cover were not analysed in this study because in the recent scenario simulations of the
CLIMSEA ensemble sea-ice cover is systematically underestimated. However, we found that future sea-ice cover
is projected to be considerably reduced, with an on-average ice-free Bothnian Sea and western Gulf of Finland.
Recent results by Höglund et al. (2017) confirmed earlier results by Meier (2002b) and Meier et al. (2011d; 2014),
see BACC Author Team (2008).

The various scenario simulation sets have in common that plausible nutrient input changes have a bigger impact
on changes in biogeochemical variables, such as nutrient, phytoplankton and oxygen concentrations, than of either
the projected changes in climate, such as warming, or changes in vertical stratification. The latter would be caused
by increased freshwater inputs, SLR or changes in regional wind fields, assuming RCP4.5 or RCP8.5 scenarios.
Long-term simulations of past climate support these results. Although historical warming had an impact on the
size of the present-day hypoxic area, model results suggest that hypoxia in the Baltic Sea is best explained by the
increases in nutrient inputs due to population growth and intensified agriculture since 1950 (Gustafsson et al.,
2012; Carstensen et al., 2014; Meier et al., 2012a; 2019c, d). Hypoxia is also a feature of the Medieval Climate
Anomaly (Zillén and Conley, 2010). However, a preliminary attempt to simulate the past 1000 years could not
explain the low-oxygen conditions without substantial increases in nutrient inputs (Schimanke et al., 2012). Thus,
the sensitivity of state-of-the-art physical-biogeochemical models to various drivers can be questioned and it is
clear that the models do not reproduce all important processes.

As outlined in previous assessments, current and future bioavailable nutrient inputs from land and atmosphere are
unknown and were consequently classified as one of the largest uncertainties (Meier et al., 2019b). For a more
detailed discussion of uncertainties in Baltic Sea projections, the reader is referred to Meier et al. (2018a; 2019b;
825 2021).

**5 Summary**
As shown in Section 3, the latest published scenario simulations confirm the findings of the first and second
assessments of climate change in the Baltic Sea region (BACC Author Team, 2008; BACC II Author Team, 2015),
namely that, in all projections driven by RCP4.5 and RCP8.5 and by four selected ESMs of CMIP5, water
temperature is projected to increase and sea-ice cover to decrease significantly. In the two RCP scenarios, the
ensemble mean annual changes in SST between 1978–2007 and 2069–2098 are 2°C and 3°C respectively.
Warming would enhance the stability across the seasonal thermocline and cause a shallower MLD during summer.
During winter, however, the mixed layer in the northern Baltic Sea would be deeper, probably because of the
declining sea-ice cover and the associated intensification of wind speed, waves and vertical mixing. Both the
frequency and the duration of MHWs would increase significantly, in particular south of 60°N and in the coastal
zone (except in regions with frequent upwellings).

The spatial patterns of seasonal SST trends projected for 2006–2099 are similar to those of historical
reconstructions of the period 1850–2008, although in most regions the magnitude of the simulated trends is larger.
The largest trends are those in summer in the northern Baltic Sea (Bothnian Sea and Bothnian Bay) and thus in
regions where under a warmer climate sea ice would melt earlier or disappear completely due to the ice-albedo
feedback. This implies that, with increasing warming, SST trends in the northern Baltic Sea will become larger
than those in the southern Baltic Sea. Accordingly, in contrast to the present climate, in which mean SSTs
considerably decline from south to north, in a future climate the north-south temperature gradient will weaken.

In contrast to previous scenario simulations, recent scenario simulations considered the impact of the global mean
SLR on Baltic Sea salinity, which for the ensemble mean salinity would more or less completely compensate for
the effects of the projected increasing river runoff. However, as future changes in all three drivers of salinity (wind,
runoff and SLR) are highly uncertain, the spread in the salinity projections of the various ESMs is larger than any
signal.

In agreement with earlier assessments, we conclude that SLR has a greater potential to increase surge levels in the
Baltic Sea than does changing wind speed or direction. For the latter, there have been no statistically significant
changes during the 21$^{st}$ century thus far.

In agreement with earlier studies, changes in nutrient input according to the BSAP or REF scenarios will have a
larger impact on biogeochemical cycling in the Baltic Sea than will a changing climate driven by RCP4.5 or
RCP8.5 scenarios. Furthermore, the impact of climate change will be more pronounced under higher than under
lower nutrient conditions. Hence, without further nutrient input reductions, as suggested by the BSAP,
eutrophication and oxygen depletion will worsen. However, the response determined in recent studies differs
considerably from the responses reported in previous studies, because of more plausible assumptions regarding
historical and future nutrient inputs. In some cases this has led to opposite signs in the response of bottom oxygen
concentrations. The new scenarios suggest that implementation of the BSAP would lead to a significant
improvement in the ecological status of the Baltic Sea regardless of the applied RCP scenario.

However, recent studies identified SLR as a new global driver. Depending on the combination of SLR and RCP
scenarios, the impact on the bottom oxygen concentration may be significant. A higher mean sea level relative to
the seabed at the sills would cause increased saltwater inflows, a stronger vertical stratification in the Baltic Sea
and a larger hypoxic area. The relationship between vertical stratification and the size of the hypoxic area was
confirmed in historical measurements. Nevertheless, recent studies suggest that the difference in future nutrient
emissions between the BSAP and REF scenarios is a more important driver than the projected changes in climate
with respect to changes in hypoxic area, phytoplankton concentration, water transparency (Secchi depth), primary
production and nitrogen fixation.

The currently available ensembles of scenario simulations are larger than in previous studies. Consequently, the
uncertainty range covered by the assessed ESMs and, in turn, the spread of the results are also larger. However,
the ensemble size might still be too small and model uncertainty is very likely underestimated. Moreover, natural
variability might be a more important source of uncertainty than previously considered for applications in the
Baltic Sea.

In the present climate, the climate variability of the Baltic Sea region during winter is dominated by the impact of
the NAO. However, in the past the correlation between the NAO and regional variables such as water temperature
or sea ice varied in time. The low-frequency changes in this correlation are projected to continue. Furthermore,
systematic changes in the influence of the large-scale atmospheric circulation on regional climate and on the NAO
itself could not be detected. While a northward shift in the mean summer position of the westerlies at the end of
the 21st century compared to the 20th century was reported (Gröger et al., 2019), it was based upon a limited set
of simulations with a few ESMs.
**Acknowledgements**
During the preparation of this paper, shortly before its submission, our co-author, Christian Dieterich, passed away
(1964–2021). This sad event marked the end of the life of a distinguished oceanographer and climate scientist who
made important contributions to climate modelling for the Baltic Sea, North Sea and North Atlantic regions.
This study belongs to the series of Baltic Earth Assessment Reports (BEARs) of the Baltic Earth Program (Earth
System Science for the Baltic Sea Region) and is dedicated to Christian Dieterich. The work was financed by the
Copernicus Marine Environment Monitoring Service through the CLIMSEA project (Regionally downscaled
climate projections for the Baltic and North seas, CMEMS 66-SE-CALL2: LOT4) and by the Swedish Research
Council for Environment, Agricultural Sciences and Spatial Planning (Formas) through the ClimeMarine project
within the framework of the National Research Programme for Climate (grant no. 2017-01949). Regional climate
scenario simulations were conducted on the Linux clusters Krypton, Bi, Triolith and Tetralith, all operated by the
National Supercomputer Centre in Sweden (NSC, http://www.nsc.liu.se/). Resources on Triolith and Tetralith were
funded by the Swedish National Infrastructure for Computing (SNIC) (grants SNIC 002/12-25, SNIC 2018/3-280
and SNIC 2019/3-356). Furthermore, we thank Berit Recklebe (Leibniz Institute for Baltic Sea Research
Warnemünde, IOW) for technical support and Dr. Boris Chubarenko and Dr. Vladimir Ryabchenko for very good
comments that helped to improve an earlier version of the manuscript.


**Figures**

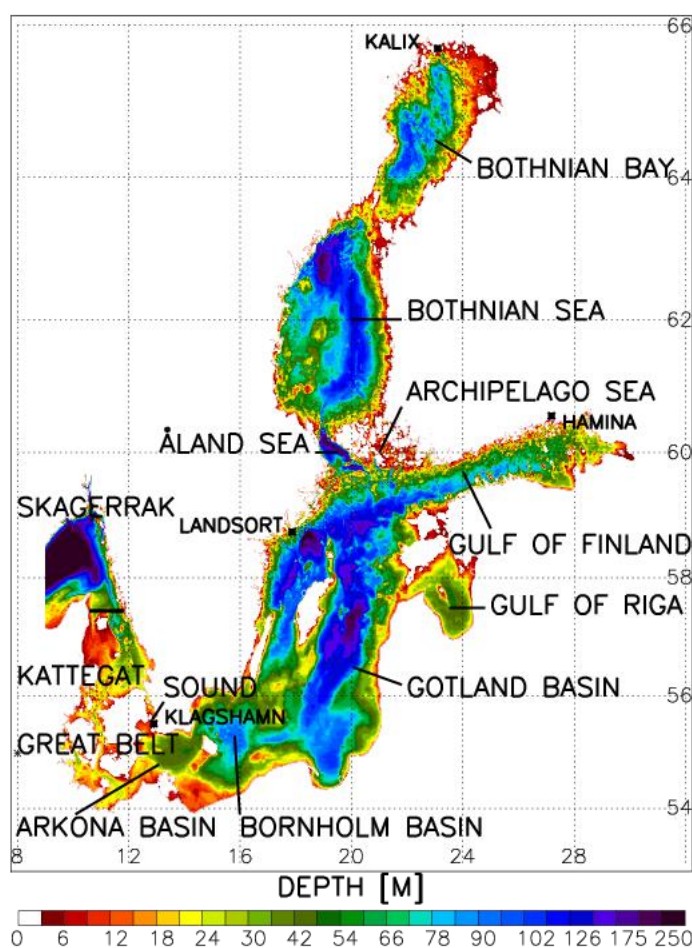


**Figure 1:** Bottom topography of the Baltic Sea (depth in m). The Baltic proper comprises the Arkona Basin,
Bornholm Basin and Gotland Basin. The border of the analysed domain of the Baltic Sea models is shown as a
black line in the northern Kattegat. The tide gauges Klagshamn (55.522ºN, 12.894ºE), Landsort (58.742ºN,
17.865ºE), Hamina (60.563ºN, 27.179ºE), and Kalix (65.697ºN, 23.096ºE) are also depicted.


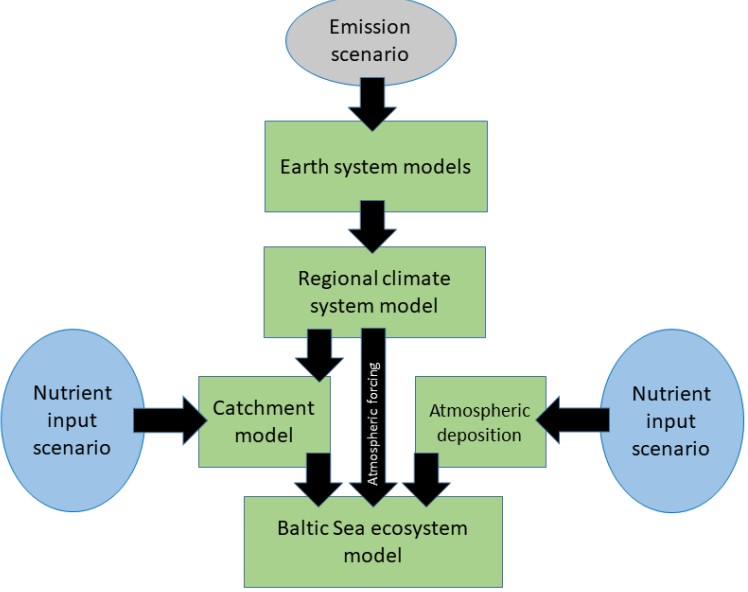


**Figure 2.** Dynamical downscaling approach for the Baltic Sea region. The models for the various components of
the Earth System are explained in Section 2. (Source: Meier et al., 2021)


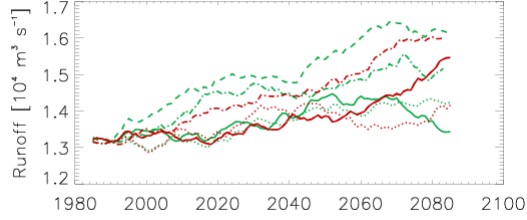


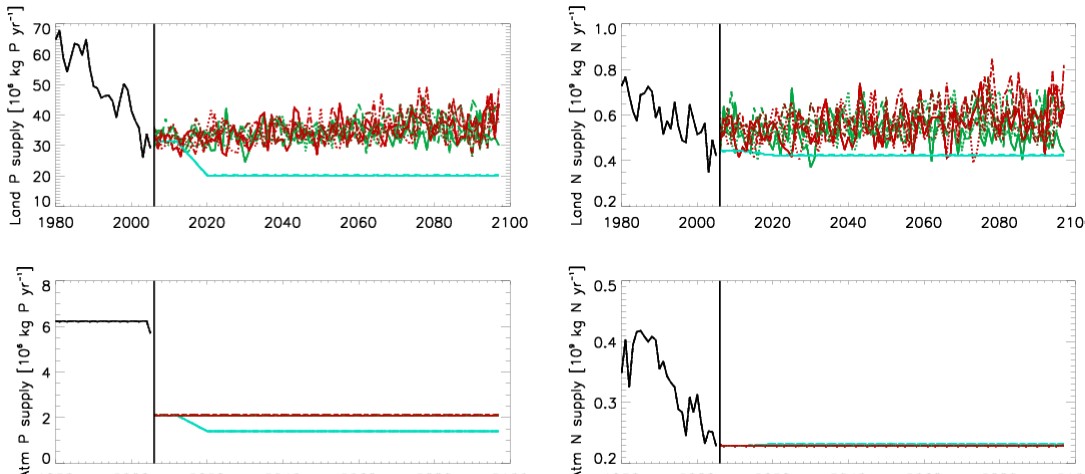



**Figure 3.** Projections of river discharge and nutrient inputs from land and atmosphere into the entire Baltic Sea according to the BalticAPP and CLIMSEA scenario simulations. Upper panel: Low-pass filtered runoff data (in $m^3\ s^{-1}$) using a cut-off period of 30 years in four regionalised Earth System models (ESMs; illustrated by different line types) under RCP4.5 (green) and RCP8.5 (red) scenarios. Lower panels: Bioavailable phosphorus (in $10^6$ kg P year$^{-1}$, left panels) and nitrogen inputs (in $10^9$ kg N year$^{-1}$, right panels) from land (upper panels) and the atmosphere (lower panels) under RCP4.5, BSAP (blue), RCP4.5, REF (green), RCP8.5, BSAP (orange) and RCP8.5, REF (red) scenarios. Nutrient inputs during the historical period are depicted in black. The nutrient input scenario WORST of the BalticAPP scenario simulations (Saraiva et al., 2019a; their Fig. 4) is not displayed, neither are the ECOSUPPORT nutrient input scenarios (Gustafsson et al., 2011; their Fig. 3.1). (Source: Meier et al., 2021)

933

934

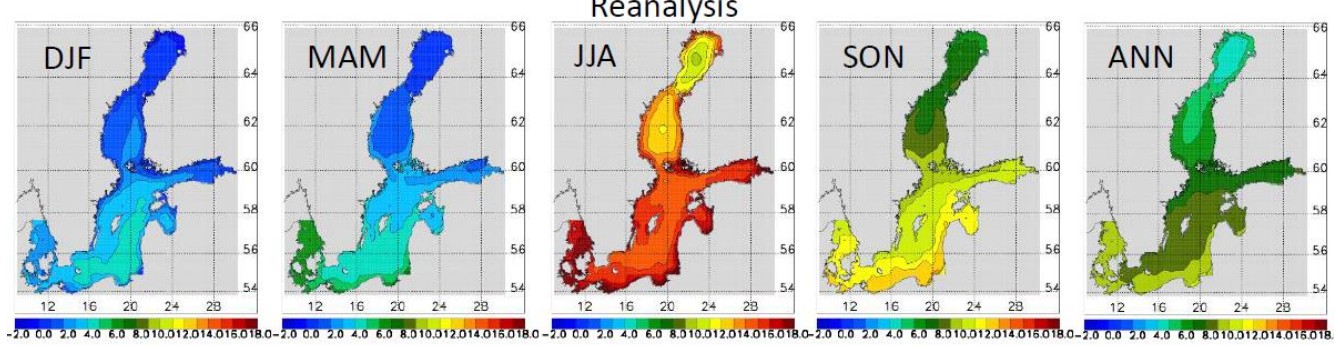

935

**Figure 4:** Upper panels: Annual and seasonal mean sea surface temperature (SST, in °C) in a reanalysis of data from 1970 to 1999 (Liu et al., 2017). Lower panels: Difference between the climatologies of the ensemble mean of the regionalised ESMs used in BalticAPP (Saraiva et al., 2019a) and CLIMSEA (Meier et al., 2021) during the historical period (1976-2005) and those of the reanalysis data. From the left to right: winter (December–February, DJF), spring (March–May, MAM), summer (June–August, JJA), autumn (September–November, SON) and annual (ANN) mean SSTs or SST differences.


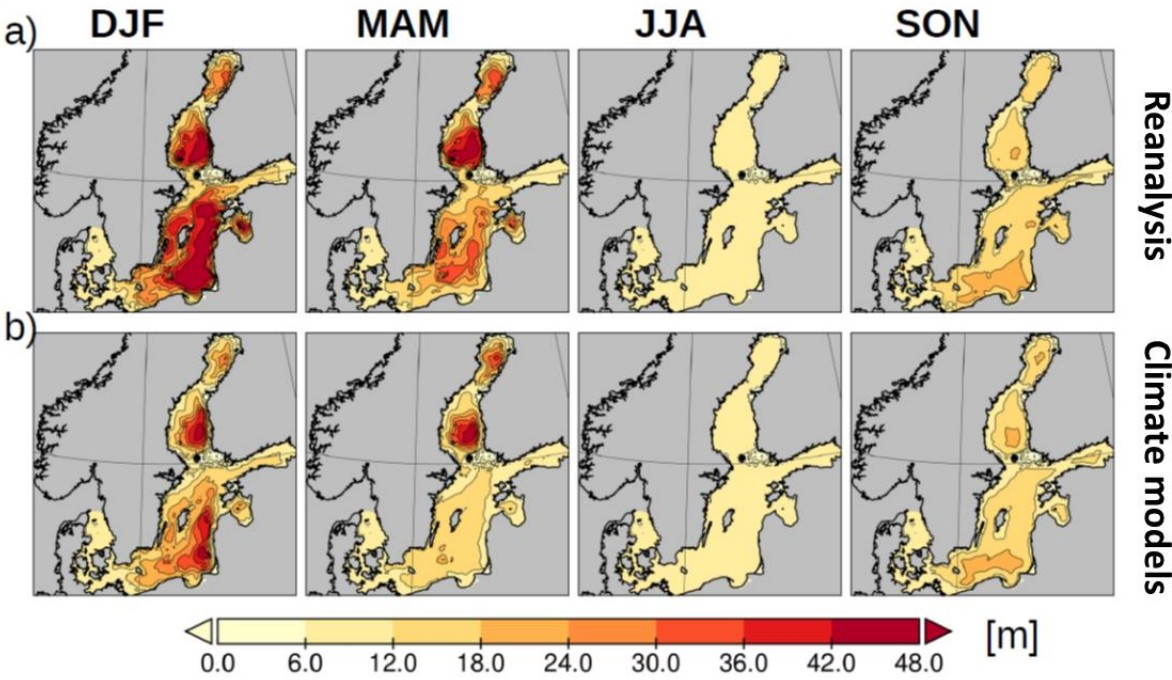


**Figure 5:** Mixed-layer thickness calculated according to the criterion following de Boyer Montégut et al. (2004).

a) Reanalysis data (Liu et al., 2017). b) Ensemble mean over the four models (Saraiva et al., 2019a). Shown are

the averages during 1976–1999.


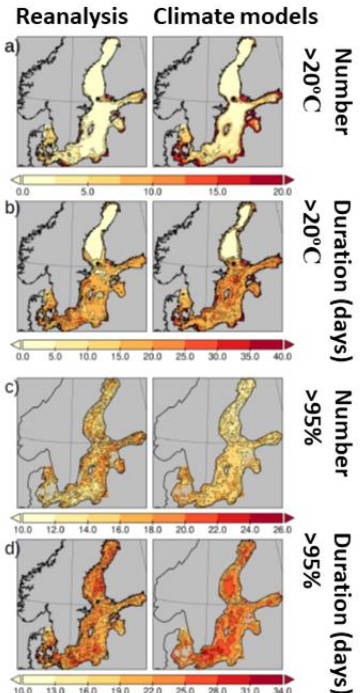


**Figure 6:** a) Number of >10-day periods in which the SST exceeds 20°C. b) Average duration of the periods displayed in a). c) Number of 10-day periods in which the SST exceeds the 95th percentile. d) Average duration of the periods displayed in c). Left column: reanalysis data (Liu et al., 2017). Right column: ensemble mean of the scenario simulations driven by four ESMs (Saraiva et al., 2019a). The analysis period is 1976–1999. Note the different colour scales used in c) and d).

954

955

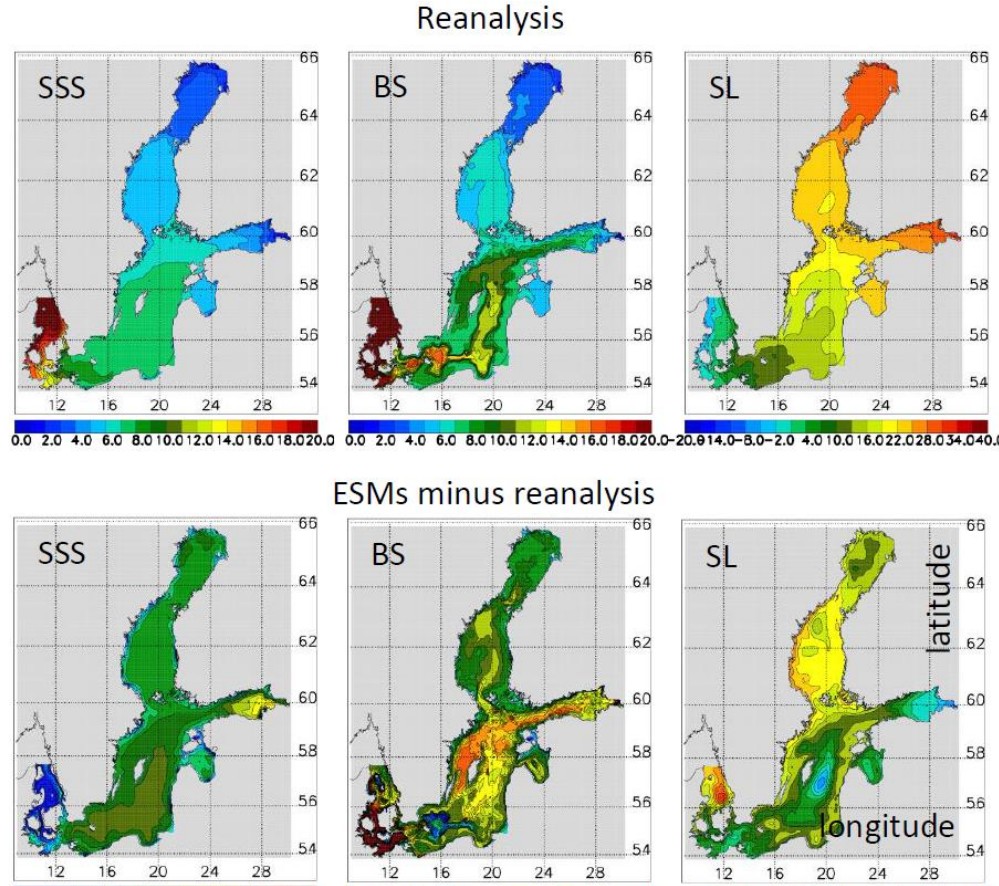

956

**Figure 7:** Upper panels: Annual mean sea surface salinity (SSS) and bottom salinity (BS) (in g kg⁻¹) and the winter (December–February) mean sea level (SL; in cm) in the reanalysis data of 1971–1999 (Liu et al., 2017; from left to right). Note that the model results for sea level are given in the Nordic height system 1960 (NH60) by Ekman and Mäkinen (1996). Lower panels: Difference between the climatologies of the ensemble mean of the regionalised ESMs used in BalticAPP (Saraiva et al., 2019a) during the historical period (1976–2005) and those of the reanalysis data.

963

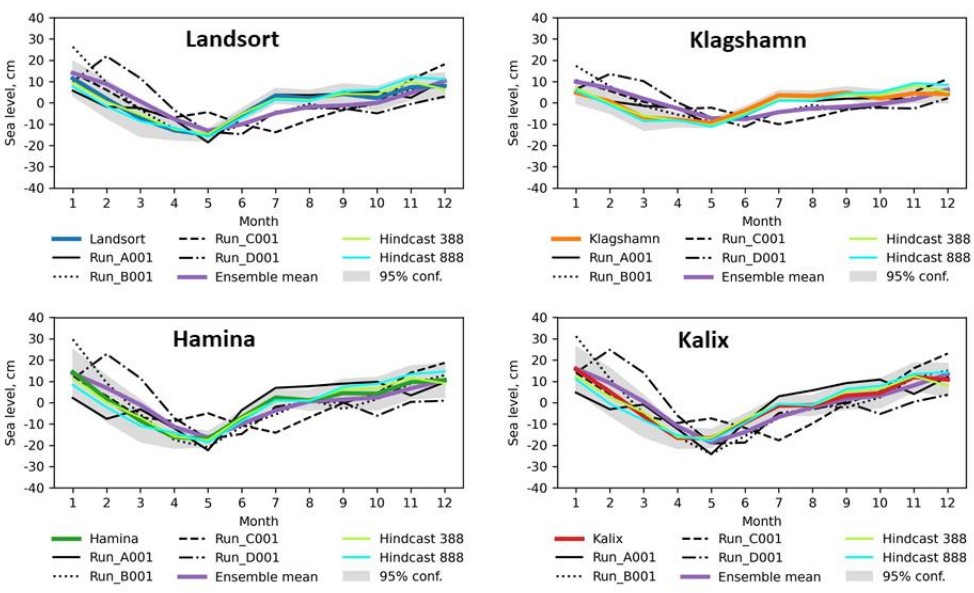

**Figure 8:** Monthly mean sea level according to a hindcast (driven by a regionalised reanalysis of atmospheric surface fields, i.e. RCA4 driven by ERA-40; Hindcast 388), reanalysis with the data assimilation of Liu et al. (2017) (Hindcast 888) and four climate simulations following Saraiva et al. (2019a) (Run_A001, …, Run_D001), the ensemble mean and observations for the historical period 1976–2005 at the sea level stations Klagshamn, Landsort, Hamina and Kalix (for the locations, see Figure 1). The 95% confidence interval of the observations is shown as a grey shaded area.

972

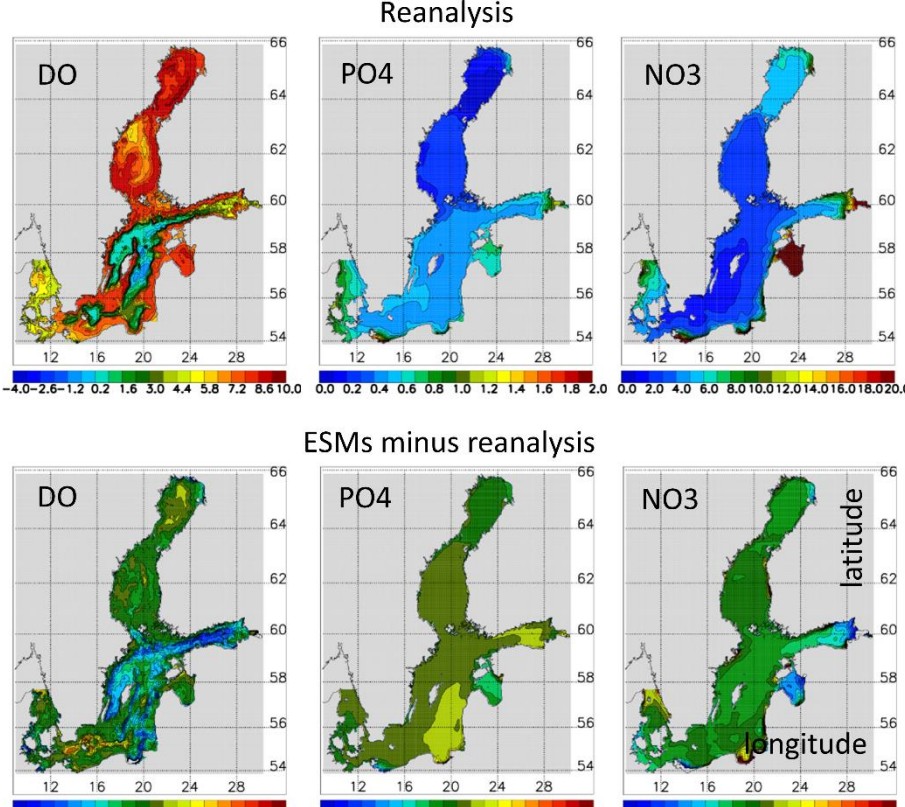

973

**Figure 9:** Upper panels: Summer (June–August) mean bottom dissolved oxygen (DO) concentrations (in mL L$^{-1}$),
winter (December–February) mean surface phosphate (PO4) concentrations (in mmol P m$^{-3}$) and winter
(December–February) mean surface nitrate (NO3) concentrations (in mmol N m$^{-3}$) in the reanalysis data of 1976–
1999 (Liu et al., 2017). Negative oxygen concentration equivalents denote hydrogen sulphide concentrations, with
1 mL H$_2$S L$^{-1}$ = –2 mL O$_2$ L$^{-1}$. Nutrient concentrations are vertically averaged for the upper 10 m. Lower panels:
Difference between the climatologies of the ensemble mean of the ESMs (Saraiva et al., 2019a) and those of the
reanalysis data of the historical period (1976–2005).

981

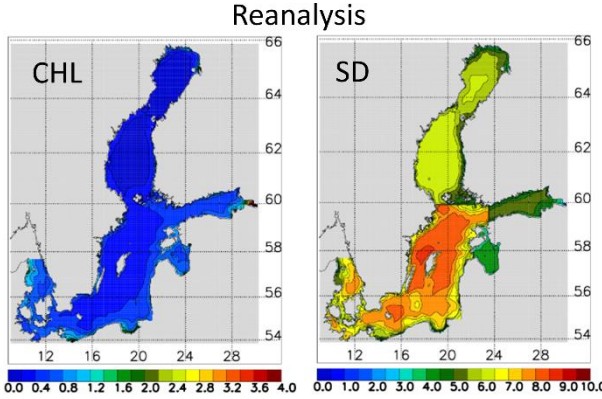

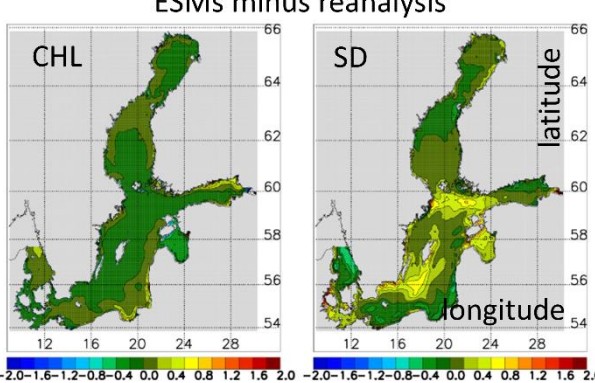

**Figure 10:** Upper panels: Annual mean phytoplankton concentrations (CHL; in mg Chl m$^{-3}$) and annual mean Secchi depth (SD; in cm) of the reanalysis data for 1976–1999 (Liu et al., 2017). Phytoplankton concentrations are vertically averaged for the upper 10 m. Since in the calculation of the Secchi depth as background only one value for the concentration of yellow substances per sub-basin is available, artificial borders between sub-basins become visible. Lower panels: Difference between the climatologies of the ensemble mean of the ESMs (Saraiva et al., 2019a) and those of the reanalysis data for the historical period (1976–2005).

990

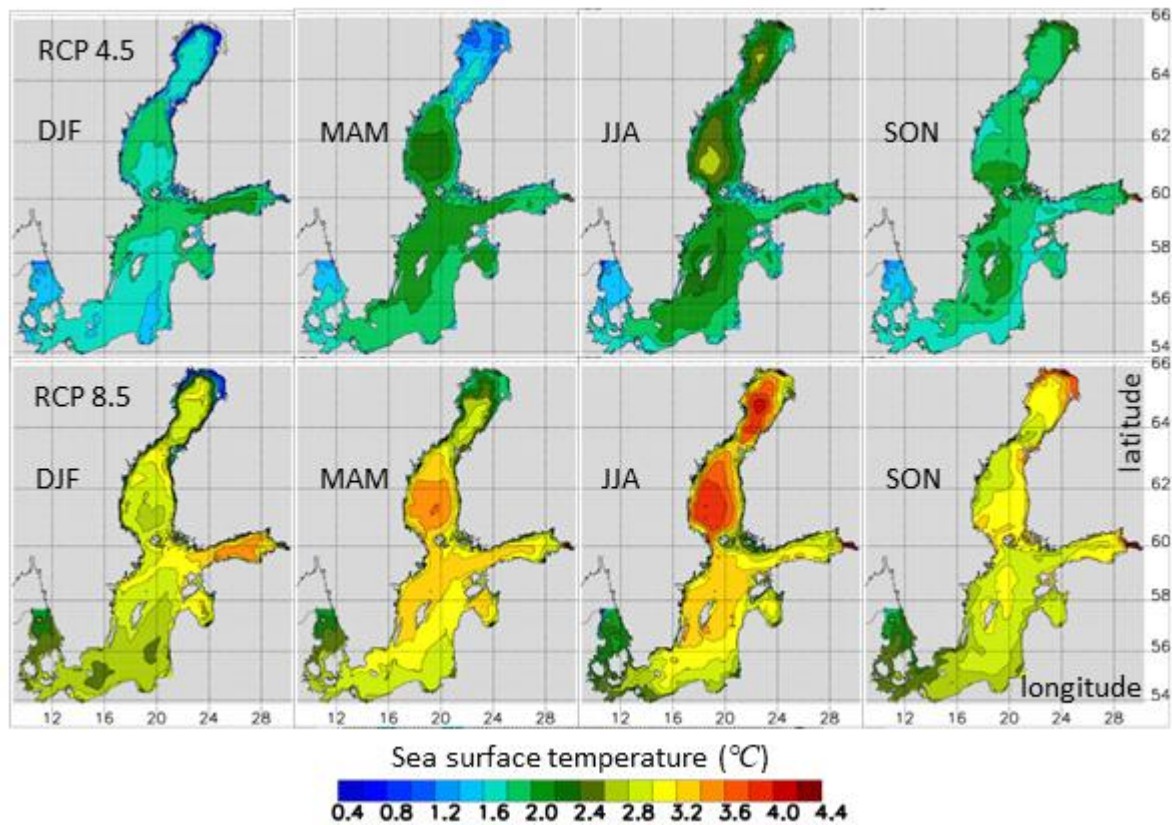

991

**Figure 11.** Changes in seasonal mean SST as simulated by the CLIMSEA ensemble (Meier et al., 2021). From
left to right, mean SST changes (in °C) in winter (December, January and February; DJF), spring (March, April
and May; MAM), summer (June, July and August; JJA) and autumn (September, October and November; SON)
between 1976–2005 and 2069–2098 under RCP4.5 (upper panels) and RCP8.5 (lower panels).

996

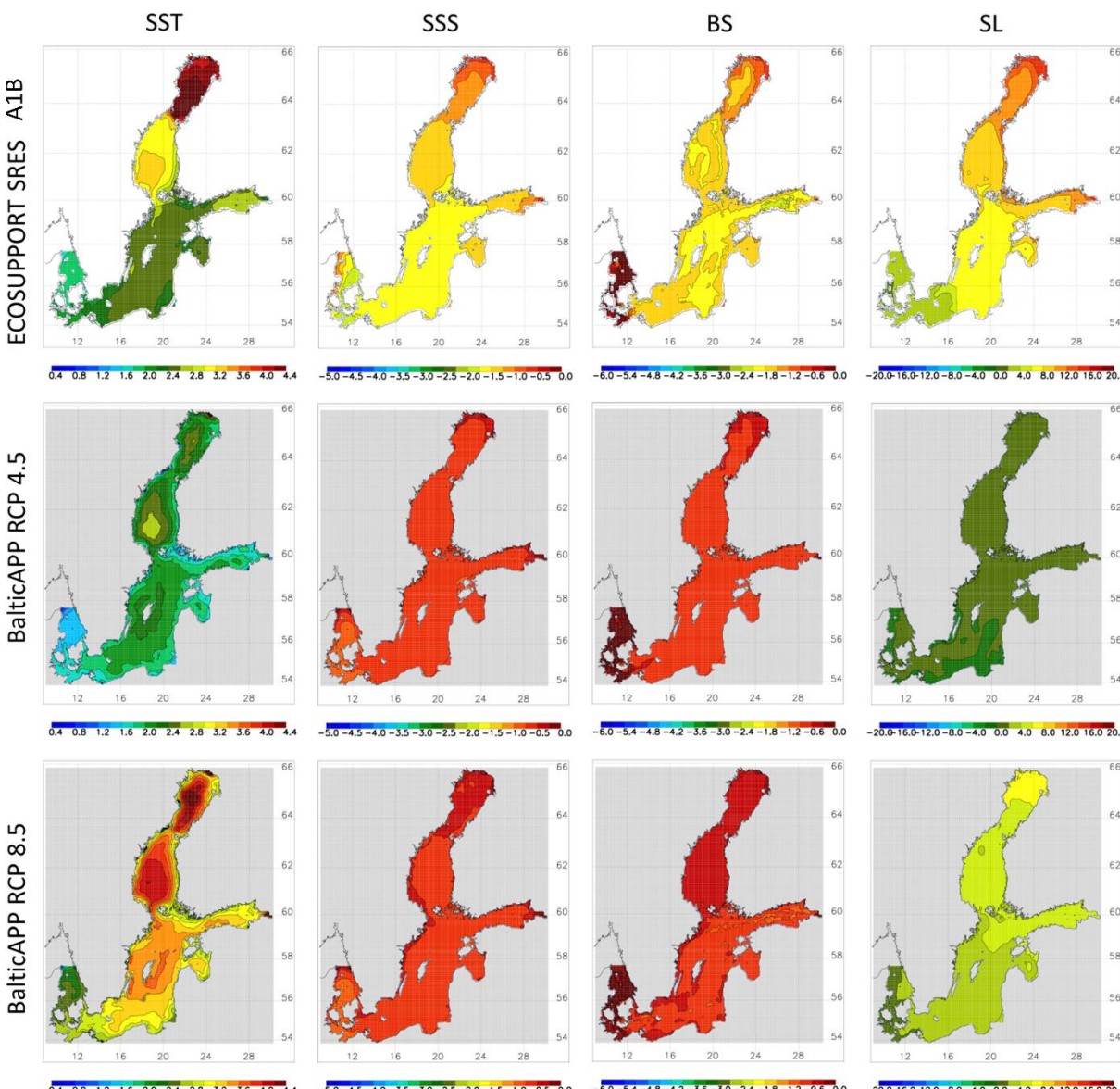

997

**Figure 12:** From left to right, changes in the mean SST (°C) in summer (June–August), the annual mean sea surface salinity (SSS; g kg⁻¹), annual mean bottom salinity (BS; g kg⁻¹), and winter (December–February) mean sea level (SL; cm) between 1978–2007 and 2069–2098. From top to bottom, the results of the ensembles ECOSUPPORT (white background, Meier et al., 2011b), BalticAPP RCP4.5 (grey background, Saraiva et al., 2019a) and BalticAPP RCP8.5 (grey background, Saraiva et al., 2019a).

1003

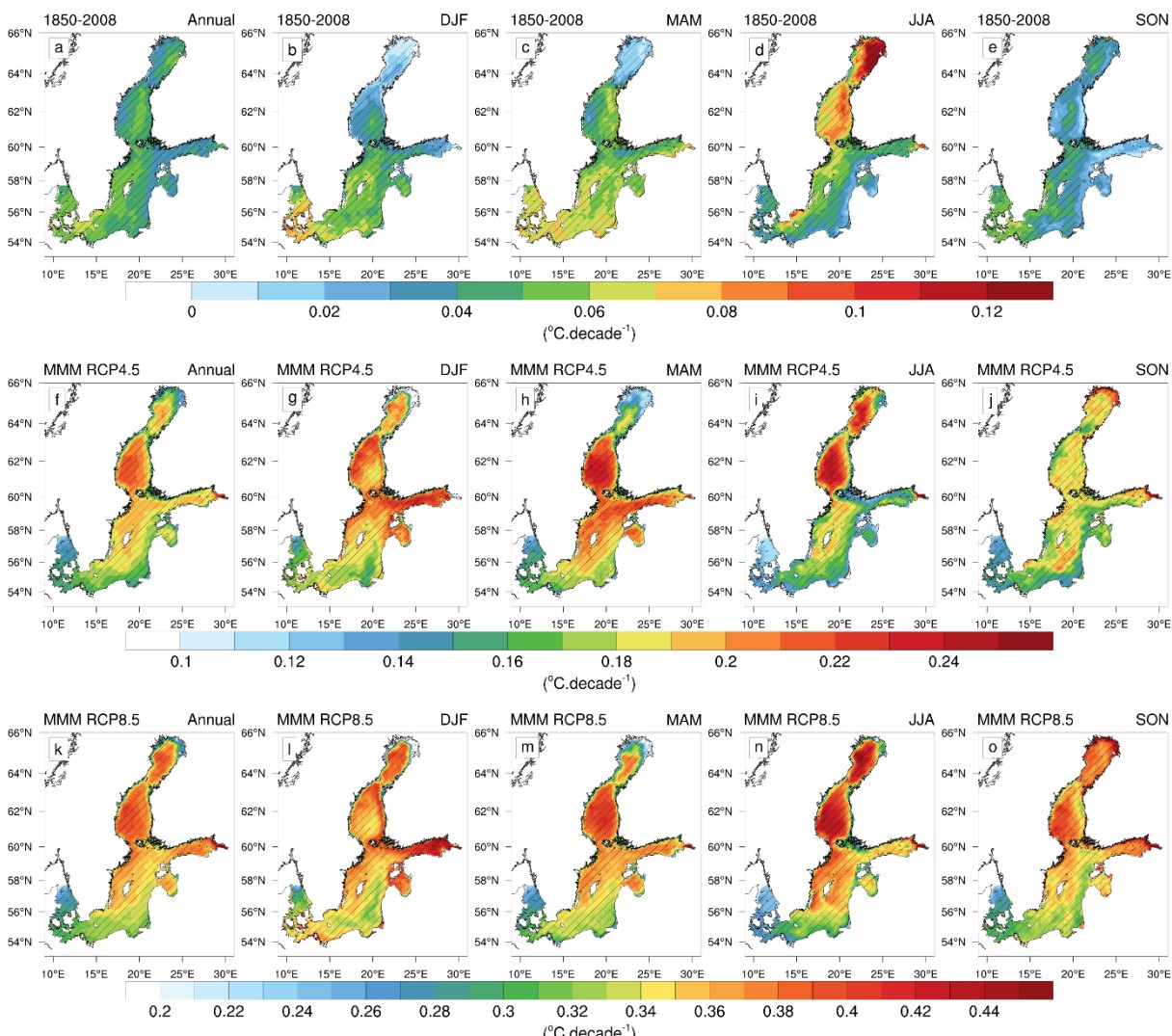

1004

**Figure 13**: Multi-model mean (MMM) of annual (a, f, k) and seasonal (b–e, g–j, l–o) SST trends (in °C decade⁻¹) computed for the period 1850–2008 (top), 2006–2099 in RCP4.5 (middle) and RCP8.5 (bottom) scenario. Hatched areas represent the regions where the trend is statistically significant (p < 0.05, Mann-Kendall test). Data for historical reconstructions and projections are from Meier et al. (2019d) and Saraiva et al. (2019a) respectively.


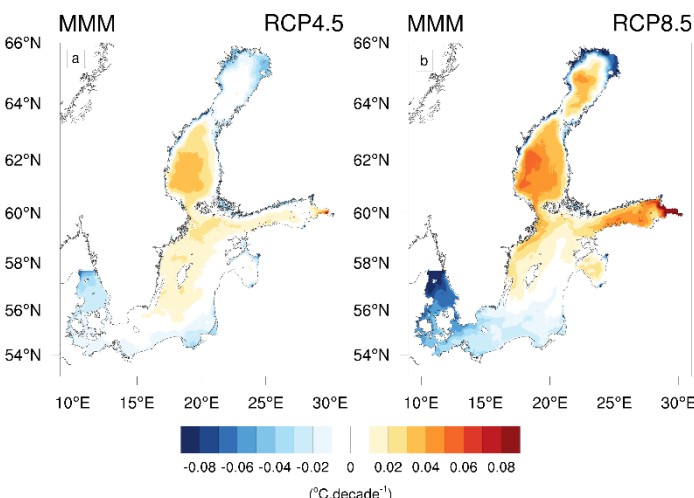


**Figure 14:** Multi-model mean (MMM) of the annual SST trends relative to the spatial average (in °C decade$^{-1}$)
for a) RCP4.5 and b) RCP8.5 scenario simulations. (Data source: Saraiva et al., 2019a)


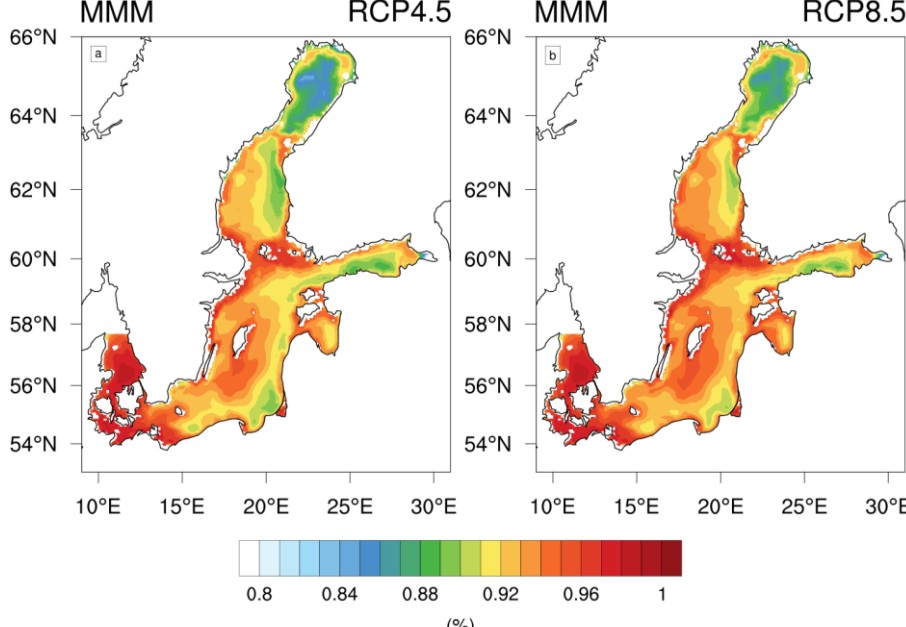

**Figure 15:** Multi-model mean (MMM) explained variance (in percent) between the monthly mean SST and the
forcing air temperature over the period 2006–2099 in the a) RCP4.5 and b) RCP8.5 scenario simulations. (Data
source: Saraiva et al., 2019a)


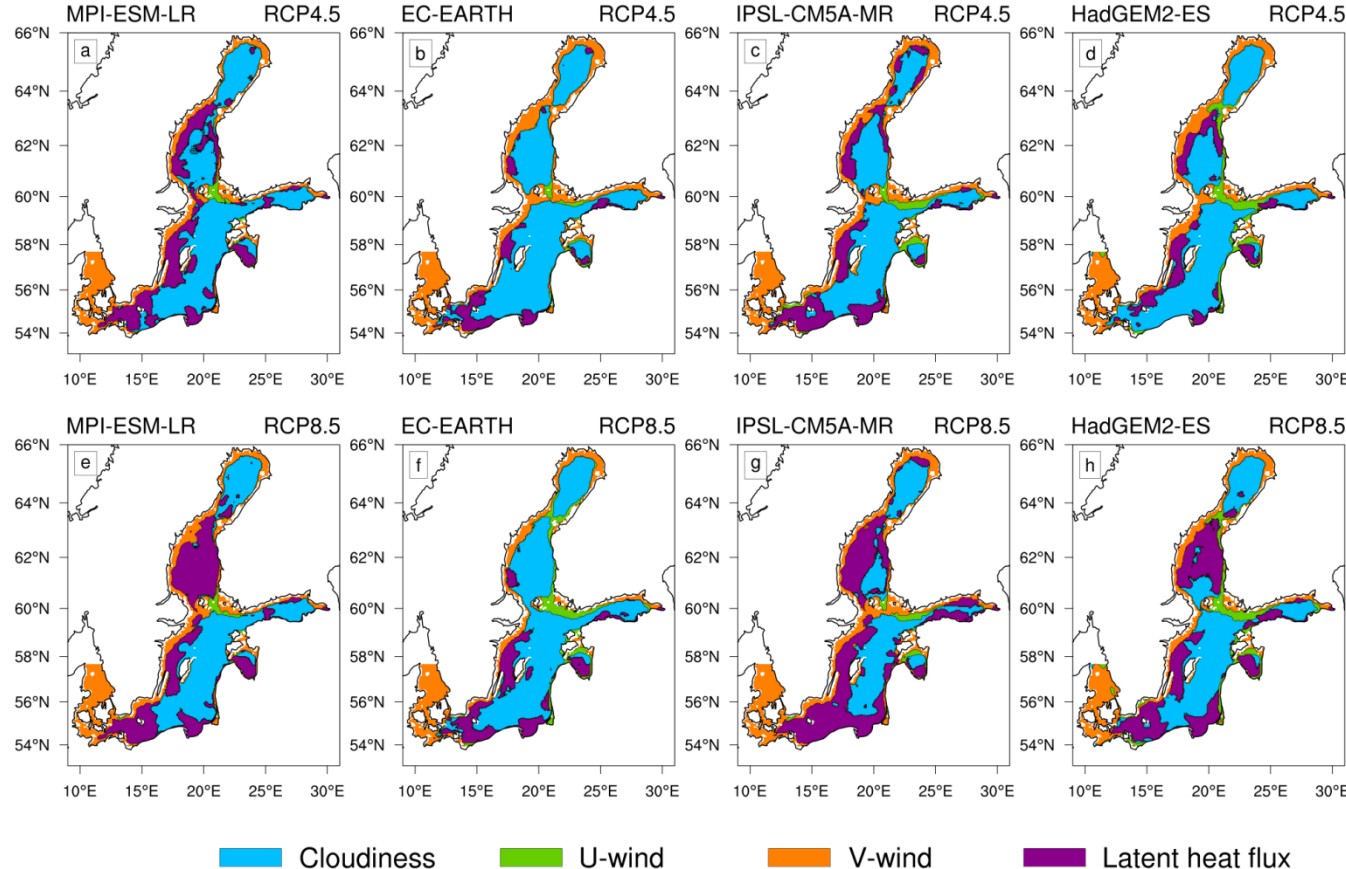

**Figure 16:** Results of the cross-correlation analysis of the detrended SST (monthly mean) with the wind
components, latent heat flux and cloudiness. Maps of the atmospheric drivers with the highest cross-correlations
in the RCP4.5 (top) and RCP8.5 (bottom) scenarios for various GCM forcings (Saraiva et al., 2019a). From left to
right: MPI-ESM-LR, EC-EARTH, IPSL-CM5A-MR, HadGEM2-ES.

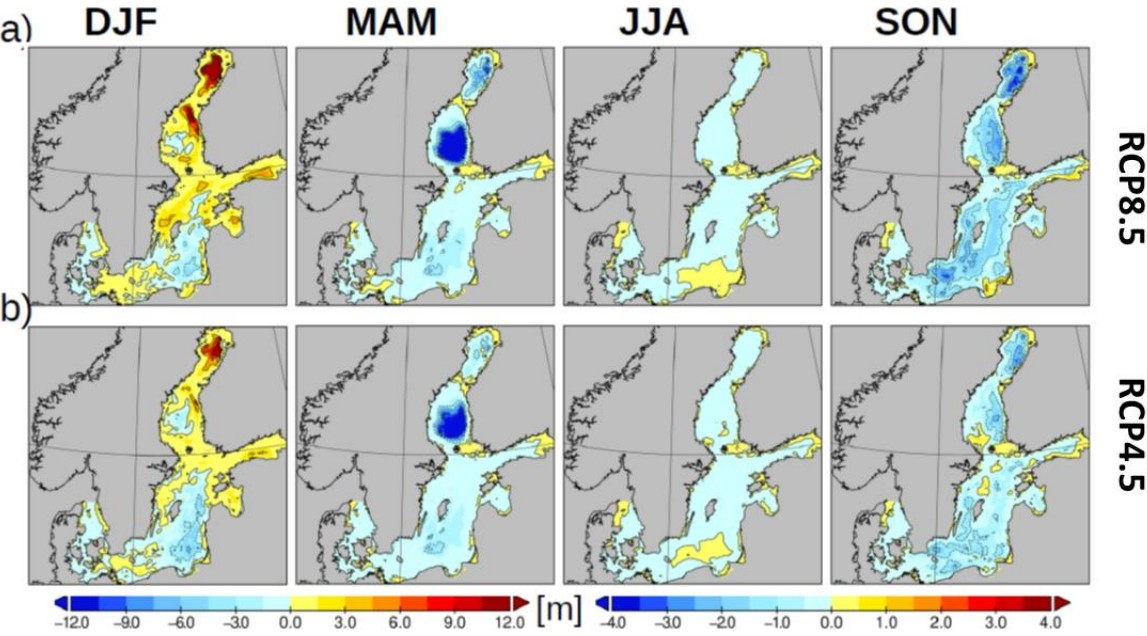


**Figure 17.** Mixed-layer depth calculated according to the criterion of de Boyer Montégut et al. (2004). Shown are
the ensemble average changes of four different ESMs between 1976–2005 and 2069–2098 with the mean sea level
rises a) 0.90 m (RCP8.5) and b) 0.54 m (RCP4.5). (Data source: Meier et al., 2021)

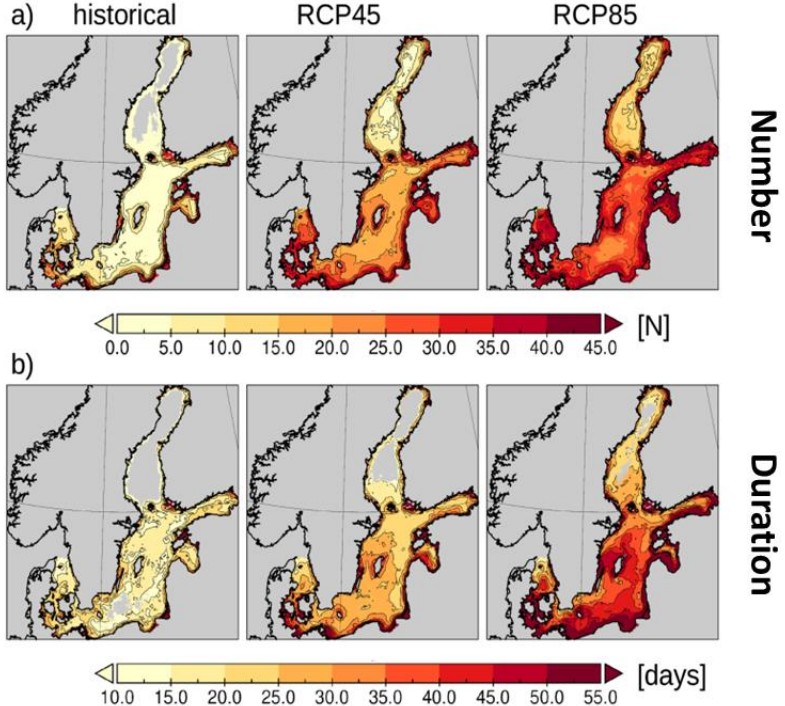


**Figure 18.** a) Number of heat waves (defined as the number of periods ≥ 10 days in which the water temperature is ≥ 20°C) for historical (1976–2005), and future (2069–2098) climates. b) Average duration of the heat waves. Note that no temperature bias adjustment was done prior to the analysis. Shown are the ensemble averages of four different ESMs with the mean sea level rises a) 0.54 m (RCP4.5) and b) 0.90 m (RCP8.5). (Data source: Meier et al., 2021)

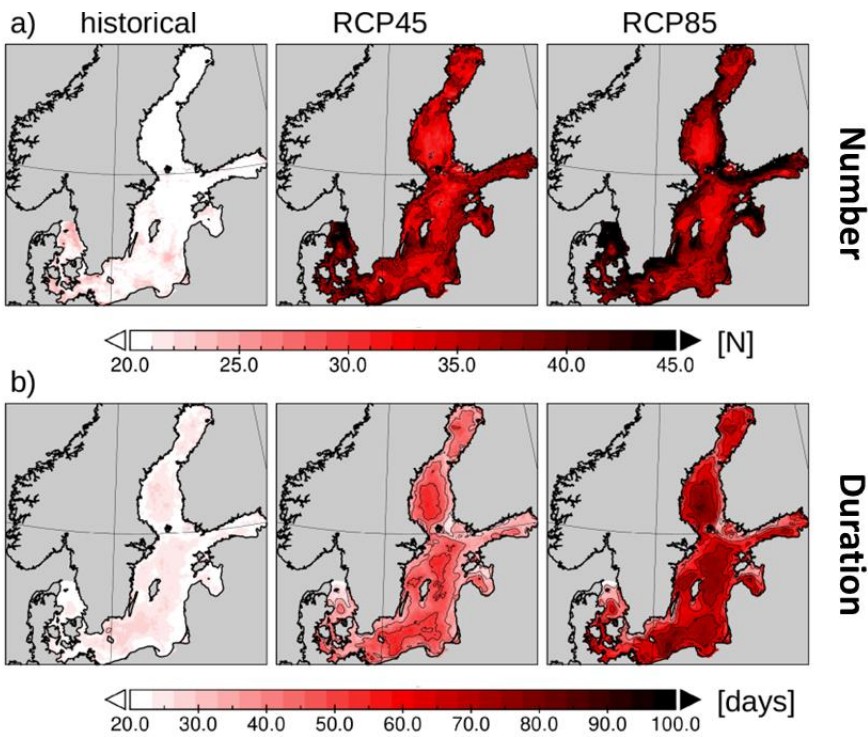

1038

**Figure 19.** Same as in Figure 18 but for heat waves defined as periods of ≥10 days in which the water temperature

1040    exceeded the 95th percentile of the historical reference temperature. (Data source: Meier et al., 2021)

1041

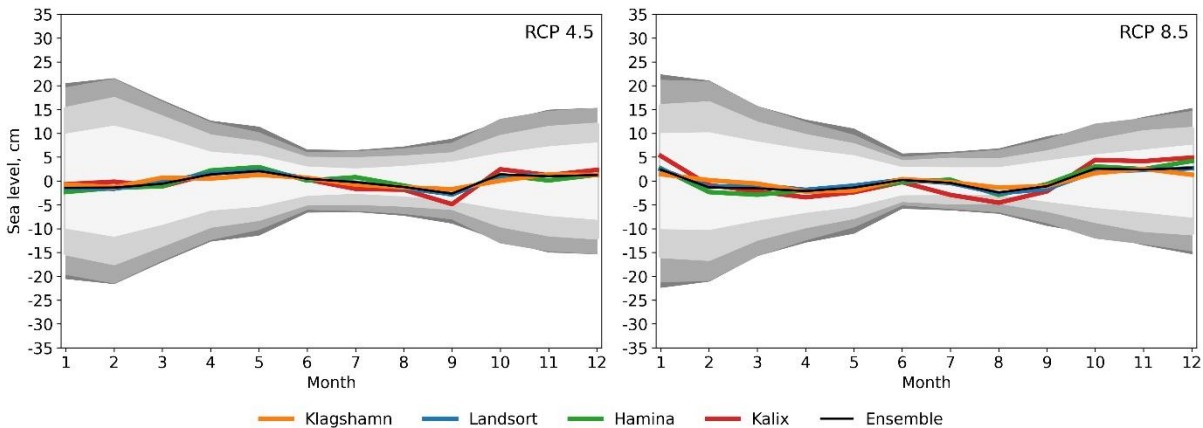

**Figure 20:** Monthly mean sea level changes between 1976–2005 and 2069–2098 at Klagshamn, Landsort, Hamina and Kalix (for the locations, see Figure 1) for RCP4.5 (left panel) and RCP8.5 (right panel). Shown are the changes relative to the mean sea level rise a) 0.54 m (RCP4.5) and b) 0.90 m (RCP8.5). Shaded areas in white to dark grey denote 99% confidence limits of internal variability at Klagshamn to Kalix respectively. The chosen model approach does not indicate any non-linear effects for scenarios with a larger rise in sea level. (Data source: Meier et al., 2021)

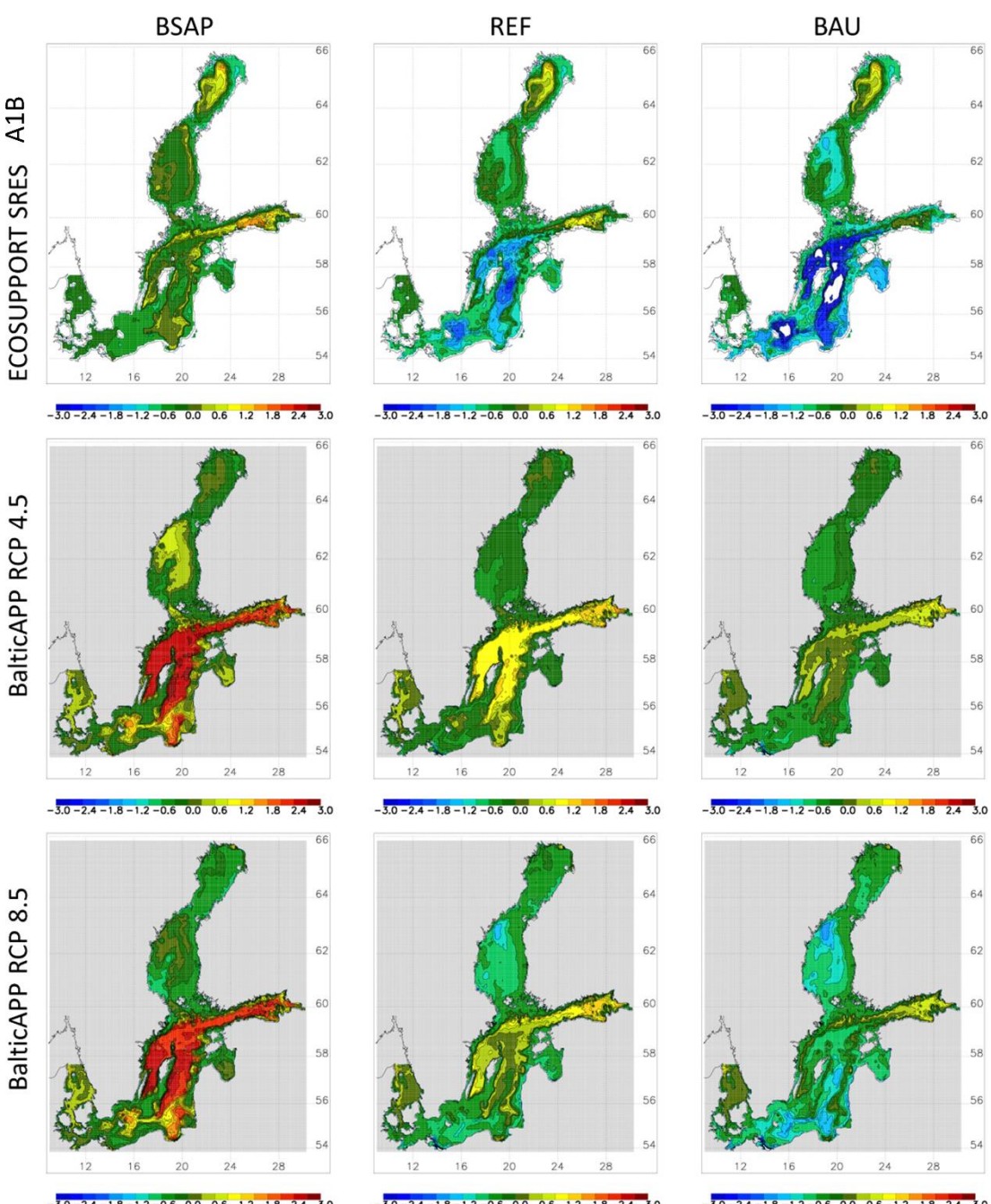

**Figure 21:** Ensemble mean changes in the bottom dissolved oxygen concentration (mL L$^{-1}$) in summer (June–August) between 1978–2007 and 2069–2098. From left to right, the results of the nutrient input scenarios of the Baltic Sea Action Plan (BSAP), Reference (REF) and Business-As-Usual (BAU). From top to bottom, the results of the ensembles ECOSUPPORT (white background; Meier et al., 2011b), BalticAPP RCP4.5 (grey background; Saraiva et al., 2019a) and BalticAPP RCP8.5 (grey background; Saraiva et al., 2019a).


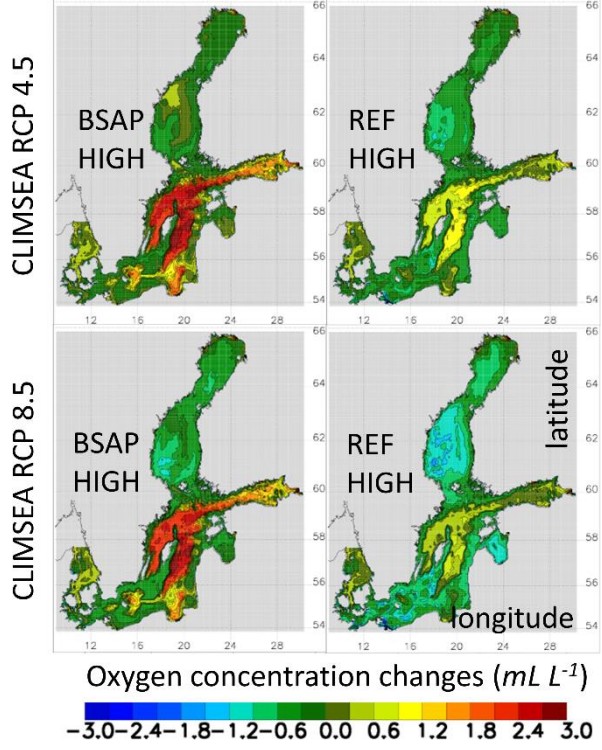

Oxygen concentration changes (*mL L⁻¹*)


**Figure 22:** As in Figure 21 but for CLIMSEA RCP4.5 (upper panels) and CLIMSEA RCP8.5 (lower panels) under
a high sea level rise scenario, i.e. 1.26 m (RCP4.5) and 2.34 m (RCP8.5). Left and right columns show the BSAP
and REF scenarios respectively. (Source: Meier et al., 2021)

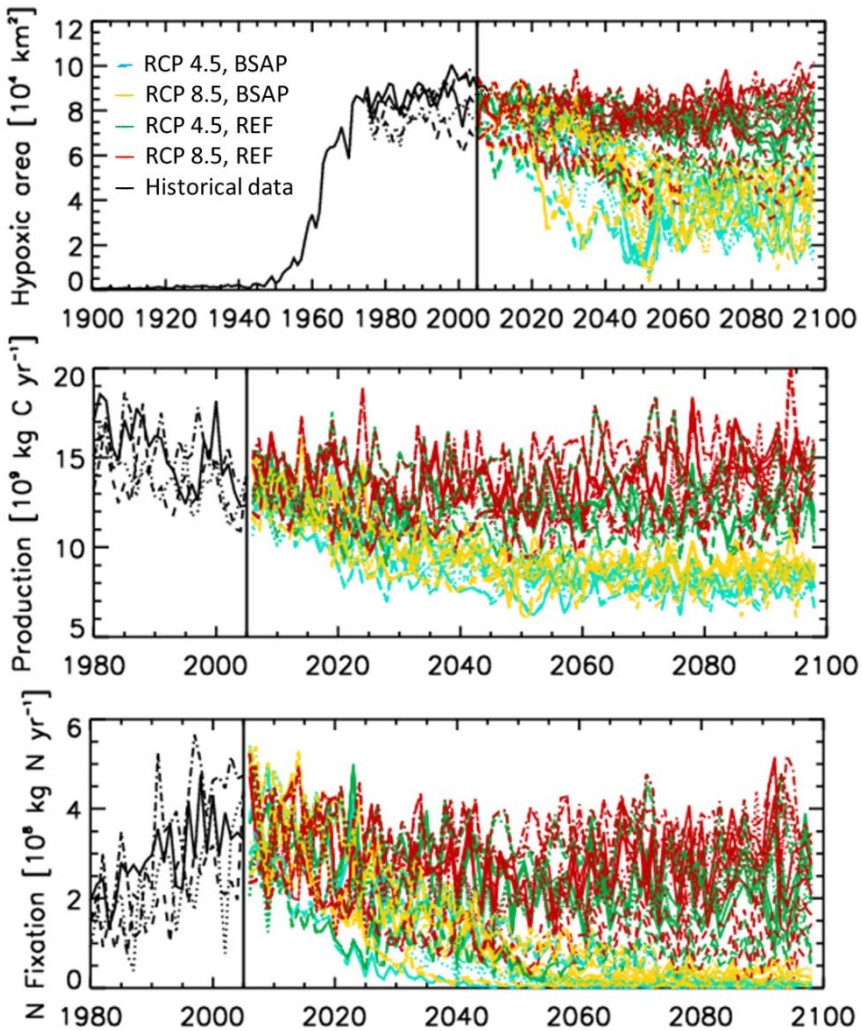

**Figure 23:** From top to bottom: hypoxic area (in km$^2$), volume-averaged primary production (in kg C year$^{-1}$) and volume-averaged nitrogen fixation (in kg N year$^{-1}$) for the entire Baltic Sea, including the Kattegat (see Figure 1) in the historical (until 2005, black lines) and scenario (after 2005, coloured lines) simulations driven by four regionalised ESMs (illustrated by different line types) under RCP4.5, BSAP (blue), RCP4.5, REF (green), RCP8.5, BSAP (orange) and RCP8.5, REF (red) scenarios. A spin-up simulation since 1850 was performed, as illustrated by the evolution of hypoxia. (Source: Meier et al., 2021)

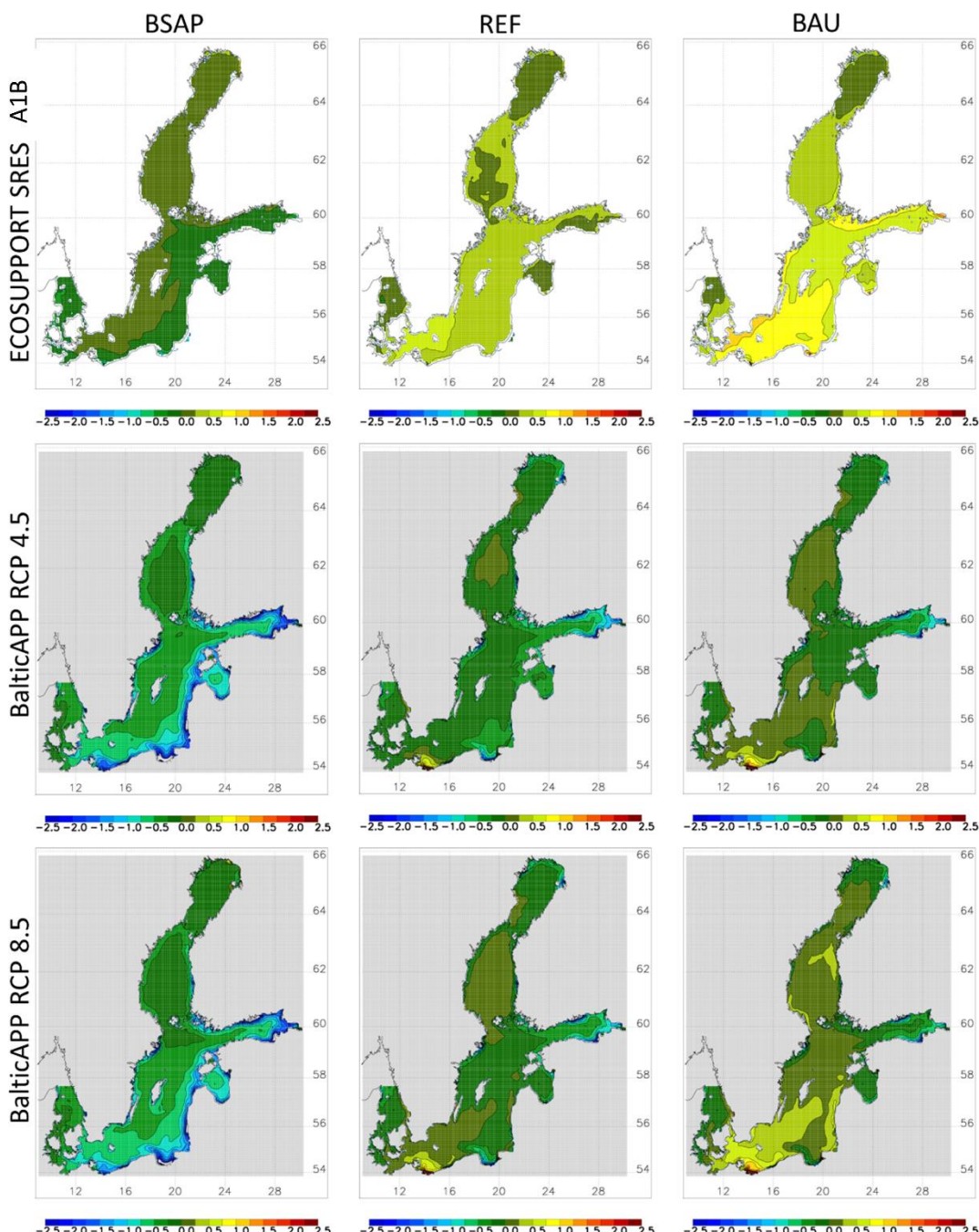


**Figure 24:** As in Figure 21 but for annual mean surface phytoplankton concentration changes (mg Chl m$^{-3}$).
Concentrations are vertically averaged for the upper 10 m. (Source: Meier et al., 2011b; Saraiva et al., 2019a)


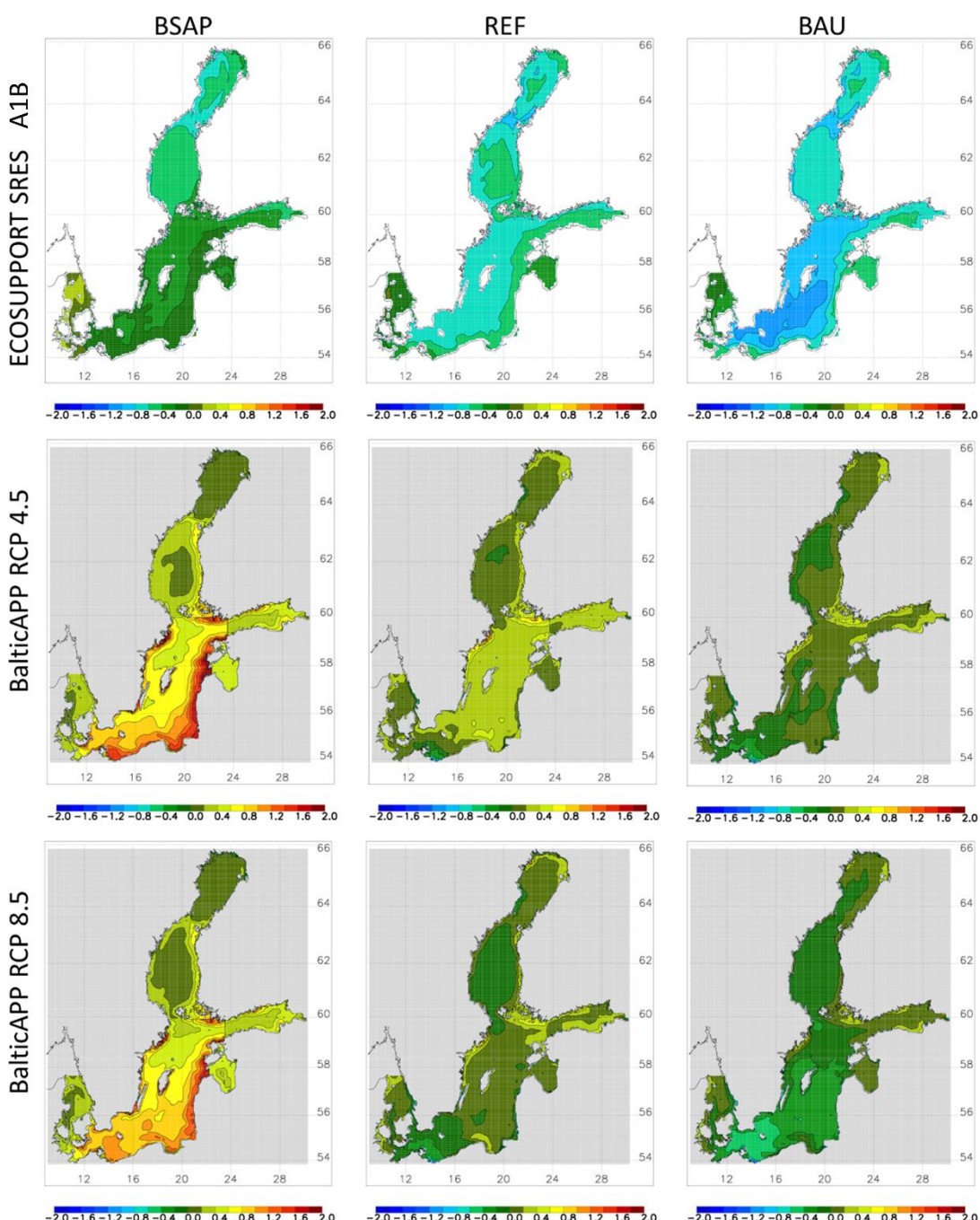


**Figure 25.** As in Figure 21 but for changes in the annual mean Secchi depth (m). (Source: Meier et al., 2011b; Saraiva et al., 2019a)


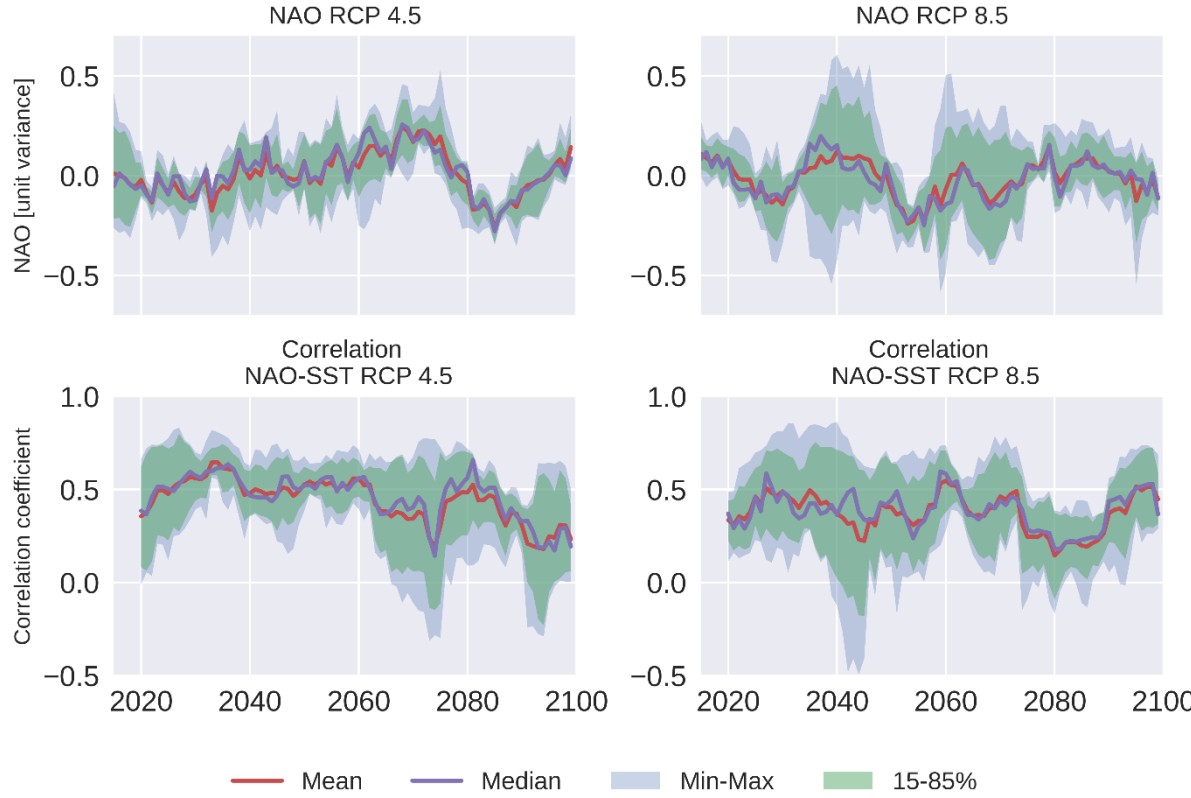

**Figure 26.** Ensemble 10-year running mean North Atlantic Oscillation (NAO) index (upper panels) and 10-year

running correlation between the NAO and area averaged SST in the Baltic Sea (lower panels) under RCP4.5 (left

panels) and RCP8.5 (right panels) scenarios. Depicted are winter (December–February) mean, median, minimum,

maximum and 15th and 85th percentiles. (Data source: Meier et al., 2021)

 **Tables**

**Table 1.** Selected ensembles of the scenario simulations for the Baltic Sea carried out in international projects (AR
= IPCC Assessment Report, GCM = General Circulation Model, RCSM = Regional Climate System Model,
RCAO = Rossby Centre Atmosphere Ocean model, RCA4 = Rossby Centre Atmosphere model Version 4, NEMO
= Nucleus for European Modelling of the Ocean, REMO = Regional Model, MPIOM = Max Planck Institute
Ocean Model, HAMSOM = Hamburg Shelf Ocean Model)

| Project | Swedish Regional Climate Modelling Program | Advanced modelling tool for scenarios of the Baltic Sea ECOsystem to SUPPORT decision making | Holocene saline water inflow changes into the Baltic Sea, ecosystem responses and future scenarios | Building predictive capability regarding the Baltic Sea organic/inorganic carbon and oxygen systems | Wellbeing from the Baltic Sea - applications combining natural science and economics | Impacts of Climate Change on Waterways and Navigation | Regionally downscaled climate projections for the Baltic and North Seas |
|---|---|---|---|---|---|---|---|
| Acronym | SWECLIM | ECOSUPPORT | INFLOW | Baltic-C | BalticAPP | KLIWAS | CLIMSEA |
| Duration | 1997-2003 | 2009-2011 | 2009-2011 | 2009-2011 | 2015-2017 | 2009-2013 | 2018-2020 |
| Project summaries | Rummukainen et al., 2004 | Meier et al., 2014 | Kotilainen et al., 2014 | Omstedt et al., 2014 | Saraiva et al., 2019a | Bülow et al., 2014 | Meier et al., 2021 |
| GCMs | AR3 | AR4 | AR4 | AR4 | AR5 | AR4/AR5 | AR5 |
| RCSM | RCAO | RCAO | RCAO | RCA | RCA4-NEMO | REMO-MPIOM, REMO-HAMSOM, RCA4-NEMO | RCA4-NEMO |
| Horizontal resolution atmosphere/ ocean | 50 km/10.8 km | 25 km/3.6 km | 25 km/3.6 km and 50 km/3.6 km for paleoclimate | 25 km /horizontally integrated | 25 km /3.6 km | varying | 25 km/3.6 km |
| Period(s) | 1961–1990 and 2071–2100 | 1961-2099 | 1961-2099 and 950-1800 AD | 1960-2100 | 1976-2100, improved initial conditions | 1961-2099 | 1976-2100 |
| Ocean model | One physical Baltic Sea model | Three physical-biogeochemical Baltic Sea models | See ECOSUPPORT | One physical-biogeochemical Baltic Sea model including the carbon cycle | One physical-biogeochemical Baltic Sea model | Two physical regional models with focus on the Baltic Sea and North Sea regions and one physical-biogeochemical ocean model | One physical-biogeochemical Baltic Sea model |
| References | Döscher and Meier, 2004; Meier et al., 2004a; Meier et al., 2004b | Meier et al., 2011b; Meier et al., 2012c; Neumann et al., 2012 | See ECOSUPPORT, Schimanke and Meier, 2016 | Omstedt et al., 2012 | Saraiva et al., 2019a, b; Meier et al., 2019b | Bülow et al., 2014; Gröger et al., 2019; Dieterich et al., 2019 | Gröger et al., 2021b; Meier et al., 2021 |


**Table 2.** Salinity projections assessed by the BACC Author Team (2008), BACC II Author Team (2015) and
BEAR (this study). Salinity changes depend on the changes in the wind field (in particular, the west wind
component), river discharge and sea level rise (SLR). The changes refer to the mean differences between historical
and future periods. (Data sources: Meier et al., 2006; 2011b; 2021)

|  | Historical period | Future period | West wind | River discharge (%) | SLR | Salinity |
|---|---|---|---|---|---|---|
| BACC (2008) | 1969–1990 | 2071–2100 | Large increase | -8 to +26 | 0 | 0 to −3.7 g kg$^{-1}$ |
| BACC II (2015) | 1978–2007 | 2069–2098 | Small increase | +15 to +22 | 0 | −1 to −2 g kg$^{-1}$ |
| BEAR (this study) | 1976–2005 | 2069–2098 | No significant change | +2 to +22 | Medium SLR +0.54 to +0.90 m | No robust change, with a considerable spread |



**Table 3.** List of scenario simulations of three ensembles. From left to right, the columns show the Earth System
Model (ESM), the Regional Climate System Model (RCSM), the Baltic Sea Ecosystem Model, the greenhouse
gas (GHG) emission or concentration scenario, the nutrient input scenario, the sea level rise (SLR) scenario and
the simulation period, including historical and scenario periods. The four nutrient input scenarios were: Baltic Sea
Action Plan (BSAP), Reference (REF), Business-As-Usual (BAU) and Worst Case (WORST). For the three SLR
scenarios in the CLIMSEA ensemble, the mean sea level changes at the end of the century are given in meters.

| ECOSUPPORT (28 scenario simulations, Meier et al., 2011b) | | | | | | |
|---|---|---|---|---|---|---|
| ESM | RCSM | Baltic Sea Model | GHG scenario | Nutrient input scenario | SLR scenario | Period |
| HadCM3 | RCAO | BALTSEM | A1B | BSAP/REF/BAU | 0 | 1961–2099 |
| ECHAM5/MPI-OM-r1 | RCAO | BALTSEM | A1B | BSAP/REF/BAU | 0 | 1961–2099 |
| ECHAM5/MPI-OM-r3 | RCAO | BALTSEM | A1B | BSAP/REF/BAU | 0 | 1961–2099 |
| ECHAM5/MPI-OM-r1 | RCAO | BALTSEM | A2 | BSAP/REF/BAU | 0 | 1961–2099 |
| HadCM3 | RCAO | MOM-ERGOM | A1B | BSAP/REF | 0 | 1961–2099 |
| ECHAM5/MPI-OM-r1 | RCAO | MOM-ERGOM | A1B | BSAP/REF | 0 | 1961–2099 |
| HadCM3 | RCAO | RCO-SCOBI | A1B | BSAP/REF/BAU | 0 | 1961–2099 |
| ECHAM5/MPI-OM-r1 | RCAO | RCO-SCOBI | A1B | BSAP/REF/BAU | 0 | 1961–2099 |
| ECHAM5/MPI-OM-r3 | RCAO | RCO-SCOBI | A1B | BSAP/REF/BAU | 0 | 1961–2099 |
| ECHAM5/MPI-OM-r1 | RCAO | RCO-SCOBI | A2 | BSAP/REF/BAU | 0 | 1961–2099 |
| BalticAPP (21 scenario simulations, Saraiva et al., 2019a) | | | | | | |
| MPI-ESM-LR | RCA4-NEMO | RCO-SCOBI | RCP4.5 | BSAP/REF/WORST | 0 | 1976–2099 |
| MPI-ESM-LR | RCA4-NEMO | RCO-SCOBI | RCP8.5 | BSAP/REF/WORST | 0 | 1976–2099 |
| EC–EARTH | RCA4-NEMO | RCO-SCOBI | RCP4.5 | BSAP/REF/WORST | 0 | 1976–2099 |
| EC-EARTH | RCA4-NEMO | RCO-SCOBI | RCP8.5 | BSAP/REF/WORST | 0 | 1976–2099 |
| IPSL-CM5A-MR | RCA4-NEMO | RCO-SCOBI | RCP4.5 | BSAP/REF/WORST | 0 | 1976–2099 |
| HadGEM2-ES | RCA4-NEMO | RCO-SCOBI | RCP4.5 | BSAP/REF/WORST | 0 | 1976–2098 |
| HadGEM2-ES | RCA4-NEMO | RCO-SCOBI | RCP8.5 | BSAP/REF/WORST | 0 | 1976–2098 |
| CLIMSEA (48 scenario simulations, Meier et al., 2021) | | | | | | |
| MPI-ESM-LR | RCA4-NEMO | RCO-SCOBI | RCP4.5 | BSAP/REF | 0/0.54/1.26 | 1976–2099 |

| MPI-ESM-LR | RCA4-NEMO | RCO-SCOBI | RCP8.5 | BSAP/REF | 0/0.90/2.34 | 1976–2099 |
|---|---|---|---|---|---|---|
| EC-EARTH | RCA4-NEMO | RCO-SCOBI | RCP4.5 | BSAP/REF | 0/0.54/1.26 | 1976–2099 |
| EC-EARTH | RCA4-NEMO | RCO-SCOBI | RCP8.5 | BSAP/REF | 0/0.90/2.34 | 1976–2099 |
| IPSL-CM5A-MR | RCA4-NEMO | RCO-SCOBI | RCP4.5 | BSAP/REF | 0/0.54/1.26 | 1976–2099 |
| IPSL-CM5A-MR | RCA4-NEMO | RCO-SCOBI | RCP8.5 | BSAP/REF | 0/0.90/2.34 | 1976–2099 |
| HadGEM2-ES | RCA4-NEMO | RCO-SCOBI | RCP4.5 | BSAP/REF | 0/0.54/1.26 | 1976–2098 |
| HadGEM2-ES | RCA4-NEMO | RCO-SCOBI | RCP8.5 | BSAP/REF | 0/0.90/2.34 | 1976–2098 |



**Table 4:** Summary of the characteristics of the ECOSUPPORT, BalticAPP and CLIMSEA scenario simulations
discussed in this study. For further details, the reader is referred to Tables 1 and 3. Acronyms are defined in
Table 5.

| Acronym | Atmospheric forcing | GHG emission or concentration scenario | Hydrological forcing | Nutrient input scenario | Special features |
|---|---|---|---|---|---|
| ECOSUPPORT | Regionalised CMIP3 data, two GCMs | SRES A1B, A2 | STAT | BSAP, REF, BAU | Three Baltic Sea models |
| BalticAPP | Regionalised CMIP5 data, four ESMs | RCP4.5, 8.5 | E-HYPE | Revised scenarios BSAP, REF, WORST | Four ESMs |
| CLIMSEA | Regionalised CMIP5 data, four ESMs | RCP4.5, 8.5 | E-HYPE | Revised scenarios BSAP, REF | SLR is considered |




**Table 5.** List of acronyms (in alphabetical order), their definitions and references

| Acronym | Definition | Comment | Reference |
|---|---|---|---|
| AMO | Atlantic Multidecadal Oscillation | Mode of climate variability | Knight et al. (2005) |
| BACC | Assessment of climate change for the Baltic Sea basin | Regional climate change assessment | BACC Author Team (2008), BACC II Author Team (2015) |
| BalticAPP | Well-being from the Baltic Sea: applications combining natural science and economics | Climate modelling project for the Baltic Sea | Saraiva et al. (2019a) |
| BEAR | Baltic Earth assessment reports | Regional climate change assessment | https://baltic.earth |
| BSAP | Baltic Sea Action Plan | Nutrient load abatement strategy for the Baltic Sea | HELCOM (2013b) |
| BSAP, REF, BAU, WORST | Baltic Sea Action Plan, Reference, Business-As-Usual and WORST | Nutrient load scenarios | Meier et al. (2012a), Saraiva et al. (2019b) |
| CLC | Cyanobacteria Life Cycle Model | Advanced biogeochemical model including a cyanobacteria life cycle | Hense and Beckmann (2006, 2010) |
| CLIMSEA | Regionally downscaled climate projections for the Baltic and North Seas | Climate modelling project for the Baltic Sea | Meier et al. (2021) |
| CMIP | Coupled Model Intercomparison Project of the World Climate Research Programme | In this study, GCM/ESM results from CMIP3 and CMIP5 were assessed | https://www.wcrp-climate.org/wgcm-cmip |
| EC-EARTH | European Countries Earth System Model | ESM, CMIP5 | https://www.knmi.nl/home |
| ECHAM5-MPI-OM | Max Planck Institute Global Climate Model | GCM, CMIP3 | Roeckner et al. (2006), Jungclaus et al. (2006) |
| ECOSUPPORT | Advanced modelling tool for scenarios of the Baltic Sea ECOsystem to SUPPORT decision making | Climate modelling project for the Baltic Sea | Meier et al. (2014) |
| E-HYPE | Hydrological Predictions For The Environment applied for Europe | Process-based multi-basin model for the land surface | https://hypeweb.smhi.se/, Arheimer et al. (2012), Hundecha et al. (2016), Donnelly et al. (2013), Donnelly et al. (2017) |

| | | | |
|---|---|---|---|
| ERA-40 | 40-year reanalysis of the European Centre for Medium Range Weather Forecast | Reanalysis data used, e.g. as atmospheric forcing for ocean models | Uppala et al. (2005) |
| ESM | Earth System Model | Model applied for global climate simulations including the carbon cycle | Heavens et al. (2013) |
| EURO-CORDEX | Coordinated Downscaling Experiment: European Domain | High-resolution climate change projections for European impact research | Jacob et al. (2014), https://euro-cordex.net/ |
| GCM | General Circulation Model | Model applied for global climate simulations | Meehl et al. (2004) |
| GHG | Greenhouse gas | Emission or concentration scenarios | Nakićenović et al. (2000), Mosset al. (2010), van Vuuren et al. (2011) |
| HadCM3 | Hadley Centre Global Climate Model | GCM, CMIP3 | Gordon et al. (2000) |
| HadGEM2-ES | Hadley Centre Global Environment Model version 2: Earth System | ESM, CMIP5 | http://www.metoffice.gov.uk |
| HELCOM | Helsinki Commission | Consists of the Baltic Sea countries and the European Union | https://helcom.fi |
| IOW | Leibniz Institute for Baltic Sea Research Warnemünde | German research institute | http://io-warnemuende.de |
| IPCC | Intergovernmental Panel of Climate Change | Generated assessment reports (AR) of past and future changes in 1990, 1995, 2001, 2008, 2013 *inter alia* based upon CMIP results | http://www.ipcc.ch |
| IPSL-CM5A-MR | Institut Pierre Simon Laplace Climate Model: Medium Resolution | ESM, CMIP5 | http://cmc.ipsl.fr/ |
| MPI-ESM-LR | Max Planck Institute Earth System Model: Low Resolution | ESM, CMIP5 | https://www.mpimet.mpg.de |
| NAO | North Atlantic Oscillation | Mode of climate variability | Hurrel (1995) |
| NOSCCA | North Sea Region Climate Change Assessment | Regional climate change assessment | Quante and Colijn (2016) |
| RCA3 | Rossby Centre Atmosphere Model version 3 | Regional climate model | Samuelsson et al. (2011) |

| | | | |
|---|---|---|---|
| RCA4-NEMO | Rossby Centre Atmosphere model Version 4: Nucleus for European Modelling of the Ocean | Coupled atmosphere-ocean model applied to the Baltic Sea and North Sea | Dieterich et al. (2013), Wang et al. (2015), Kupiainen et al. (2014), Madec (2016) |
| RCAO | Rossby Centre Atmosphere Ocean Model | Regional climate model | Döscher et al. (2002) |
| RCM | Regional Climate Model | Regional atmosphere or coupled atmosphere-ocean model applied to the dynamical downscaling of a changing climate | Giorgi (1990), Rummukainen (2010, 2016), Rummukainen et al. (2015), Feser et al. (2011), Rockel (2015), Schrum (2017) |
| RCO-SCOBI, BALTSEM, MOM-ERGOM | Model abbreviations | Coupled physical-biogeochemical models for the Baltic Sea | Meier et al. (2018a), their Tables 1 and 2 and references therein |
| RCP | Representative Concentration Pathway | Greenhouse gas concentration scenario | Moss et al. (2010), van Vuuren et al. (2011) |
| RCSM | Regional Climate System Model | Regional coupled atmosphere–sea ice–ocean–wave–land surface–atmospheric chemistry–marine ecosystem model | Giorgi and Gao (2018) |
| SRES | Special Report on Emission Scenarios | Described greenhouse gas emission scenarios, e.g. A1B, A2 | Nakićenović et al. (2000) |
| STAT | Hydrological model | Statistical model for river runoff calculated from precipitation and evaporation over land | Meier et al. (2012a) |



**Table 6.** Projected ensemble mean changes in total (land and atmosphere) bioavailable annual phosphorus (ΔP)
and nitrogen (ΔN) inputs (in ktons) into the Baltic Sea between historical (1980–2005) and future (2072–2097)
climates under the scenarios REF and BSAP. (Source: Meier et al., 2018a; their Fig. 3)

| Nutrient input scenario | REF | | BSAP | |
|---|---|---|---|---|
| Nutrient input changes | ΔP | ΔN | ΔP | ΔN |
| ECOSUPPORT BALTSEM | +2 | −17 | −15 | −208 |
| ECOSUPPORT MOM-ERGOM | +1 | −15 | −8 | −180 |
| ECOSUPPORT RCO-SCOBI | +4 | +72 | −11 | −230 |
| BalticAPP/CLIMSEA RCO-SCOBI (RCP4.5) | −18 | −129 | −34 | −269 |




**Table 7.** Ensemble mean changes in sea surface temperature (SST; in °C) in the ECOSUPPORT, BalticAPP
RCP4.5, BalticAPP RCP8.5, CLIMSEA RCP4.5 and CLIMSEA RCP8.5 scenario simulations averaged for the
Baltic Sea including the Kattegat (data sources: Meier et al., 2011b; 2021; Saraiva et al., 2019a). The changes are
calculated between historical (1978–2007 in ECOSUPPORT and 1976–2005 in BalticAPP/CLIMSEA) and future
(2069–2098) periods. (DJF = December, January, February, MAM = March, April, May, JJA = June, July, August,
SON = September, October, November)

| Δ SST | Winter (DJF) | Spring (MAM) | Summer (JJA) | Autumn (SON) | Annual mean |
|---|---|---|---|---|---|
| ECOSUPPORT SRES A1B | +2.5 | +2.8 | +2.8 | +2.5 | +2.6 |
| BalticAPP RCP4.5 | +1.7 | +1.9 | +2.0 | +1.8 | +1.8 |
| BalticAPP RCP8.5 | +2.9 | +3.2 | +3.3 | +3.0 | +3.1 |
| CLIMSEA RCP4.5 | +1.7 | +1.9 | +2.0 | +1.9 | +1.9 |
| CLIMSEA RCP8.5 | +2.8 | +3.0 | +3.0 | +2.9 | +2.9 |



**Table 8.** Ensemble mean changes in annual mean sea surface salinity (SSS; in g kg$^{-1}$), annual mean bottom salinity
(BS; in g kg$^{-1}$) and winter mean sea level (SL) relative to the global mean SL (in cm) in ECOSUPPORT, BalticAPP
RCP4.5, BalticAPP RCP8.5, CLIMSEA RCP4.5 and CLIMSEA RCP8.5 scenario simulations averaged for the
Baltic Sea including the Kattegat. For CLIMSEA, both the ensemble mean and the high SL scenarios are listed.
In ECOSUPPORT and BalticAPP/CLIMSEA, the changes between 1978–2007 and 2069–2098 and between
1976–2005 and 2069–2098 were calculated respectively. (Data sources: Meier et al., 2011b; 2021; Saraiva et al.,
2019a)

| Annual/winter changes | ECOSUPPORT A1B/A2 | BalticAPP RCP4.5 | BalticAPP RCP8.5 | CLIMSEA RCP4.5 mean | CLIMSEA RCP4.5 high | CLIMSEA RCP8.5 mean | CLIMSEA RCP8.5 high |
|---|---|---|---|---|---|---|---|
| Δ SSS | −1.5 | −0.7 | −0.6 | −0.3 | +0.2 | −0.2 | +0.6 |
| Δ BS | −1.6 | −0.6 | −0.6 | −0.0 | +0.6 | −0.0 | +1.1 |
| Δ SL | +5.5 | +0.4 | +3.7 | +0.2 | +0.1 | +3.4 | +3.2 |



**Table 9**. Salinity changes averaged for the Baltic Sea in 1988–2007 relative to 1850 as a function of sea level rise (SLR). In the reference simulation, the mean salinity is 7.42 g kg$^{-1}$. In method 1, the increase in the water level was added to the first vertical grid box of the RCO-SCOBI model, while in method 2 the increase in the water level was evenly divided between the first and second grid boxes.

| SLR (in m) | −0.24 | +0.5 | +1.0 | +1.5 (method 1) | +1.5 (method 2) |
|---|---|---|---|---|---|
| Salinity (in g kg$^{-1}$) | -0.35 | +0.71 | +1.41 | +2.10 | +2.03 |

**Table 10.** As in Table 8, but showing the ensemble mean changes in the summer mean bottom oxygen concentration (in mL L$^{-1}$) in ECOSUPPORT, BalticAPP RCP4.5, BalticAPP RCP8.5, CLIMSEA RCP4.5 and CLIMSEA RCP8.5 scenario simulations averaged for the Baltic Sea including the Kattegat. The projected changes depend on the nutrient input scenario: Baltic Sea Action Plan (BSAP), Reference (REF), Business-As-Usual (BAU) or Worst Case (WORST). (Data sources: Meier et al., 2011b; 2021; Saraiva et al., 2019a)

| Summer changes | ECOSUPPORT A1B/A2 | BalticAPP RCP4.5 | BalticAPP RCP8.5 | CLIMSEA RCP4.5 mean | CLIMSEA RCP4.5 high | CLIMSEA RCP8.5 mean | CLIMSEA RCP8.5 high |
|---|---|---|---|---|---|---|---|
| BSAP | −0.1 | +0.6 | +0.5 | +0.6 | +0.5 | +0.4 | +0.3 |
| REF | −0.6 | +0.1 | −0.2 | +0.0 | −0.1 | −0.2 | −0.4 |
| BAU | −1.1 | - | - | - | - | - | - |
| WORST | - | −0.1 | −0.5 | - | - | - | - |

**Table 11.** As in Table 8, but showing the ensemble mean changes in the annual Secchi depth (in m) in the ECOSUPPORT, BalticAPP RCP4.5, BalticAPP RCP8.5, CLIMSEA RCP4.5 and CLIMSEA RCP8.5 scenario simulations averaged for the Baltic Sea including the Kattegat. The projected changes depend on the nutrient input scenario: BSAP, REF, BAU or WORST. (Data sources: Meier et al., 2011b; 2021; Saraiva et al., 2019a)

| Annual changes | ECOSUPPORT A1B/A2 | BalticAPP RCP4.5 | BalticAPP RCP8.5 | CLIMSEA RCP4.5 mean | CLIMSEA RCP4.5 high | CLIMSEA RCP8.5 mean | CLIMSEA RCP8.5 high |
|---|---|---|---|---|---|---|---|
| BSAP | −0.3 | +0.6 | +0.6 | +0.6 | +0.6 | +0.6 | +0.6 |
| REF | −0.6 | +0.2 | 0.1 | +0.2 | +0.2 | +0.1 | +0.1 |
| BAU | −0.8 | - | - | - | - | - | - |
| WORST | - | 0.1 | −0.1 | - | - | - | - |

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
