# Peer review of "Oceanographic regional climate projections for the Baltic Sea"

_Earth System Dynamics, 2021_

## Author Comment (AC1)

**Answers to reviewer no. 1 (Dr. Boris Chubarenko) in red**

Thank you very much for the thorough review and good comments. We will follow your suggestions and will revise the manuscript accordingly

**General comments**

The paper is very informative and well structured. It is written in clear language and wishes to explain the main aspects and details to a reader sincerely.

The paper clear overviews the results of several profoundly advanced (at its time) attempts to develop the climate projections for the Baltic Sea. These attempts were made during BACC (2008), BACC II (2015). BEAR (this study), ECOSUPPORT, BalticApp and CLEAMSEA projects. All of them are mentioned in the text everywhere. It would be good to present the general overview scheme to help the reader quickly understand the differences between these initiatives and projects without reading the whole text.

A general overview is provided by Table 1. We will add a sentence at the end of the introduction section clarifying that an overview and summary is provided by Table 1.

The paper emphasizes a significant step of CLEMASEAS - including a new driver, the global sea-level rise. It appeared (according to the Conclusions) that this third driver causes "a more or less complete compensation for the projected increasing river runoff" that changes previously concluded the future drop of the average salinity. It would be better to clarify whether it is the same for all used scenarios of the global sea-level rise (0.9, 1.26 or 2.34 m water depth rise) or the extreme only?

We will better explain that we refer to the ensemble mean of the entire ensemble instead of individual simulations.

For such complicated issues as climate projections for the Baltic Sea, I would expect a more extended summary that includes final statements for all analyzed variables. Or at least the reference to the appropriate section of the text.

We will add the reference to the appropriate section.

My opinion is that the idea that NAO well controls the interannual variability of the climate variables in the Baltic Sea has outlived its usefulness. Figure 25 clearly illustrates that correlation is so low that the discussion of this relationship is on the verge of physical meaning (no technically reliable instrument will not work with such characteristic correlation). The idea to find the good simple predictor for the Baltic Sea climate variability is attractable, but unfortunately, it seems, it is not realizable.

We agree completely. This was the motivation to add Figure 25. Thank you.

It is remarkable, and I personally very much support the massage of the paper that "BSAP would lead to a significant improvement" of the state of the Baltic Sea. And, more generally, human activity in the Baltic Sea catchment and the sea has a more substantial influence than the natural influence of global climate change. In this regard, it is not clear why the previously formulated (probably in ECOSUPPORT) strong sound statement "that climate change will worsen the situation in the Baltic if people do nothing" is not included in the number of conclusions.

Good suggestion. We will add a corresponding sentence to the conclusion.

**Specific comments**

The statement (lines 796-797) needs more clarification as it is not well understood. Maybe via the words that the existed natural negative south-north gradient will be partly compensated.

We will rephrase the sentence accordingly, see:

With increasing warming, SST trends in the northern Baltic Sea would get larger relative to SST trends in the southern Baltic Sea. As in present climate mean SSTs considerably decline from south to north, this gradient will be weaker in future compared to present climate. The latter might be caused by the ice-albedo feedback.

The sentence (lines 832-835) is too long. Better to split into several more simple sentences to more clearly present the idea.

We will split the sentence into three:

These low-frequency changes in correlation were projected to continue. Furthermore, systematic changes in the influence of the large-scale atmospheric circulation on regional climate and on the NAO itself could not be detected. However, a northward shift in the mean summer position of the westerlies at the end of the twenty-first century compared to the twentieth century was reported earlier (Gröger et al., 2019).

The list of abbreviations would be helpful.

We will add a list of abbreviations.

If possible (in addition to Table 1 and Table3), the table with the list and the main characteristics of RCM (RCSM) and Baltic Sea ecosystem models would be handy to understand the progress.

We will add such a table.

**Technical corrections**

Lines 774-776: It seems that the word 'model is absent after 'physical-biochemical'.

Correct. It should be "state-of-the-art physical-biogeochemical models"

Figure 25: Please, insert the legend explanations in the figure caption.

We will revise Figure 25 and the figure caption accordingly.

Table 3: Please, introduce the column titles to link with the Table caption (via numbers, for example). The acronym of scenarios (BSAP, REF and BAU) have to be explained in the caption.

We will add column titles and explain the acronyms.

Table 4: Please, introduce the sign '+' (as in the other tables) to indicate the positive changes. There is enough space in the table to have explicit column titles; for example, December-January-February (DJF), not only DJF.

We will add the sign. As the abbreviations for the seasons are explained in the figure caption we will add the corresponding season, i.e. winter, spring, …

Table 6 and 7: the last raw 'BAU/WORST': Why is only one number given for these two scenarios? Better explain in the caption.

Good point. For clarity, we will add another row to make clear that both scenarios BAU and WORST are different. BAU was only used in ECOSUPPORT while WORST was only applied in BalticAPP.

---

## Author Comment (AC2)

**Answers to reviewer no. 2 (Dr. Vladimir Ryabchenko) in red**

Thank you very much for the thorough review and good comments. We will follow your suggestions and will revise the manuscript accordingly

**General comments**

The presented manuscript analyzes and compares the results of two ensembles of scenario simulations (performed in projects ECOSUPPORT and BalticAPP / CLIMSEA ) for the Baltic Sea including marine biogeochemistry performed using different Regional Climate System Models (RCSM). In addition to different RCSMs, these projection ensembles differ in the Earth System Model (ESM) forcing, the Baltic Sea ecosystem model, the greenhouse gas (GHG) concentration scenario, the nutrient input scenario, the sea level rise (SLR) scenario and the simulation period.

Despite the uncertainties related to global and regional climate and impact models, as well as the unknown pathways of GHG and nutrient emissions, which make it very difficult to compare these two ensembles of projections, the authors obtained a number of important new results. Among them, I note: 1) global mean SLR results in a more or less complete compensation for the Baltic Sea salinity reduction for the projected increasing river runoff, 2) the global SLR can be identified as a new driver that has a strong impact on bottom oxygen concentration, 3) most noticeable are the differences in projected biogeochemical variables between ECOSUPPORT and BalticAPP / CLIMSEA ensembles. The article is undoubtedly important and interesting for ESD readers not only with the presented results, but also with a discussion of knowledge gaps and prospects for further research. The main, but, incidentally, a small drawback of the manuscript is not always a sufficiently detailed explanation of the results obtained, which is discussed below in specific comments.

Thank you for the comments. We will try to improve our explanations of the results based on your comments in the revised manuscript.

**Specific comments**

*Abstract*

In the abstract there is no description of the differences in setups in the two ensembles of projections (ECOSUPPORT and BalticAPP / CLIMSEA simulations). It would be nice to list them in accordance with Table 3.

We prefer not to add these technical information because the abstract is already relatively long (369 words). To explain the differences between the setups of earlier ensemble studies, much more text would be needed.

***Main text***

**L.114-115**

"An overview was given by (Schrum et al., 2016)…"

Reference is missing from the bibliography

Thank you. Added.

**L.193-208, 866-874 (Fig.3)**

Nutrient input scenarios in BalticAPP and CLIMSEA simulations are described using Fig. 3. The latter is of very poor quality and does not allow you to see the differences between the blue, green, orange and red curves (Fig.3, lower panels) corresponding different scenarios. The choice of colors appears to be unfortunate. In addition, plotting all the curves on one graph creates the impression of complete chaos, nothing more (although, perhaps, the authors wanted to show just that).

The ECOSUPPORT nutrient input scenarios (Gustafsson et al., 2011; their Fig. 3.1) are not displayed in the Fig. 3. At the same time, they are important (especially for the historical period) in order to better understand the differences in the projections of biogeochemical variables in the two Ñ•omparable ensembles, and it would be good to add them as a separate panel in Fig.3.

Figure 3 is taken from a recent publication in Communications Earth and Environment. It is not necessary to distinguish individual curves because only tendencies and spread are important. Some colors are not visible because curves are identical and displayed on top of each other. Thank you very much for the very good suggestion to add information about ECOSUPPORT nutrient loads. We will add an additional table for the changes in bioavailable nutrient inputs in the various scenario simulations. We will also explain the differences in the text in more detail.

**L.460-463**

The good agreement between simulated annual mean surface phytoplankton concentrations and reanalysis data everywhere (with the exception of coastal regions) is somewhat surprising. As far as the author of the review is aware, the deviations of the simulated chlorophyll concentration from the individual observations can be very large. Perhaps the agreement is due to the small number of observations and their corresponding small contribution to the results of the reanalysis?

You are completely correct. In the reanalysis by Liu et al. (2017) nutrient and oxygen concentrations are assimilated but not chlorophyll data. We will add a sentence explaining the finding.

**L. 641-658**

"**3.2.6 Oxygen concentration and hypoxic area**

***Bottom oxygen concentration***"

This subsection discusses the differences between ECOSUPPORT and BalticAPP ensembles in bottom oxygen concentration. Discussion of differences ends with the phrase:

"These results are explained by the historical nutrient input reductions and the slow response of the Baltic Sea". (L.658)

This unsatisfactory, too general explanation makes the reader think, and due to what differences in historical nutrient input this situation could have happened, and he turns to Fig. 3 and does not find there the necessary information about historical nutrient input in ECOSUPPORT (see above remark about Fig. 3). Further, an inquisitive, but already somewhat irritated reader continues reading and discovers a detailed explanation of the differences between ECOSUPPORT and BalticAPP ensembles in the biogeochemical cycling (for some reason, in the Knowledge gaps section) in lines 751-763.

In my opinion, it would be better not to test the patience of the reader and move this explanation to section 3.2.6. It would also be good to expand this explanation by indicating the difference in the initial conditions for biogeochemical variables between ECOSUPPORT and BalticAPP ensembles and how this difference affected the final results.

We agree with your comment. The text part will be moved and the differences will be better explained.

---

## Author Response (AR1)

**List of changes**

1) The language of the entire manuscript was copy-edited and technical details raised by the reviewers were addressed.

2) All comments by reviewer 1 and 2 have been addressed. There is one exception: the extension of the abstract requested by reviewer 2 was not done.

3) In particular, a general overview is provided by Table 1. We added a sentence at the end of the introduction section clarifying that an overview and summary is provided by Table 1.

4) Concerning salinity changes, we better explained that we refer to the ensemble mean of the entire ensemble instead of individual simulations.

5) In the summary, we added a reference to the result section.

6) The changes in SST gradient were better explained: With increasing warming, SST trends in the northern Baltic Sea would get larger relative to SST trends in the southern Baltic Sea. As in present climate mean SSTs considerably decline from south to north, this gradient will be weaker in future compared to present climate. The latter might be caused by the ice-albedo feedback.

7) A too long sentence was splitted: These low-frequency changes in correlation were projected to continue. Furthermore, systematic changes in the influence of the large-scale atmospheric circulation on regional climate and on the NAO itself could not be detected. However, a northward shift in the mean summer position of the westerlies at the end of the twenty-first century compared to the twentieth century was reported earlier (Gröger et al., 2019).

8) We added a list of abbreviations.

9) We added a table characterizing the involved ensembles.

10) Thank you for the comments. We will try to improve our explanations of the results based on your comments in the revised manuscript.

11) A table for the changes in bioavailable nutrient inputs in the various scenario simulations was added

12) Details of the data assimilation by Liu et al. (2017) were added.

13) The explanation for the differences between the two ensembles, ECOSUPPORT and BalticAPP, was moved from the discussion to the result section.

---

## Author Response (AR2)

**Response to the Editor's comments (in red):**

**Comments to the author**:

The authors have prepared an excellent paper and have accounted for almost all reviewer comments and where they have not, they had good arguments. The manuscript is well structured and written and in principle ready for publication. Thank you very much for your comments. We will follow your suggestions and will revise the manuscript accordingly. See our detailed answers below.

There are, however, some (minor) technical issues in the figures and tables, which should be looked at. The readability of some figures could be improved, and the display of similarly structured figures could be homogenized. In some table captions, information is missing. See the detailed cases below.

Figures:

Figures 5, 6, 11: As done in other figures with panels organized logically in rows and columns, it would be good to indicate variables for rows and columns already in the plots, as done e.g. exemplarily in Figure 12, 21a, 23, 24. This increases the initial readability of the figures very much, if the (abbreviated) variable text is not too long. This also refers to Fig. 17, 18, 19.

Figure 13: Delete "SST trend" from panels because all panels show the same.

Figure 14: same

Done. See the revised version.

I assume the resolution of the final submitted figures will be better than in this pdf. Yes, the final versions will have higher resolution.

Tables:

Table 2: Projections for which period? We added the historical and future periods.

Table 5: Should´t this table be at the end? We followed the rule that the numbering of figures and tables follow the first mentioning in the text. The editorial office may comment.

Table 7: Changes for which period? We added the historical and future periods.

Line 1177: The correct url is: https://baltic.earth Corrected.

For clarity and correctness minor language changes were performed in lines 539, 581, 591, 645, 651, 678, 701 and 841.

Markus Meier

On behalf of the co-authors